# Copy-Paste to Mitigate Large Language Model Hallucinations

**Yongchao Long**[1,2]    **Yingying Zhang**[3]    **Xianbin Wen**[1]    **Xian Wu**[3,†]    **Yuxi Zhou**[1,†]
**Shenda Hong**[2,†]

[1]Department of Computer Science, Tianjin University of Technology, Tianjin, China
[2]National Institute of Health Data Science, Peking University, Beijing, China
[3]Tencent Jarvis Lab, Shenzhen, China
[†]Corresponding author

## Abstract

While Retrieval-Augmented Generation (RAG) enables large language models (LLMs) to generate contextually grounded responses, contextual faithfulness remains challenging as LLMs may not consistently trust provided context, leading to hallucinations that undermine reliability. We observe an inverse correlation between response copying degree and context-unfaithful hallucinations on RAGTruth, suggesting higher copying degrees reduce hallucinations by fostering genuine contextual belief. We propose **Copy-Paste**, a generation paradigm that directly embeds contextual fragments to ensure faithfulness, and instantiate it through CopyPasteLLM via two-stage high-copying preference training. We design three prompting methods to enhance copying degree, demonstrating that high-copying responses achieve superior contextual faithfulness and hallucination control. These approaches enable a fully automated pipeline that transforms generated responses into high-copying preference data for training CopyPasteLLM. On FaithEval, ConFiQA and PubMedQA, CopyPasteLLM achieves best performance in both counterfactual and original contexts, remarkably with 12.2% to 24.5% accuracy improvements on FaithEval over the best baseline, while requiring only 365 training samples—*1/50th* of baseline data. To elucidate CopyPasteLLM's effectiveness, we propose the *Context-Parameter Copying Capturing* algorithm. Interestingly, this reveals that CopyPasteLLM recalibrates reliance on internal parametric knowledge rather than external knowledge during generation. All codes are available at https://github.com/longyongchao/CopyPasteLLM

## 1 Introduction

Large language models (LLMs) have brought revolutionary breakthroughs to natural language processing (Annepaka & Pakray, 2025; Qin et al., 2024), while retrieval-augmented generation (RAG) further empowers LLMs with grounded external knowledge capabilities (Fan et al., 2024; Zhao et al., 2024). However, LLMs inevitably suffer from knowledge conflicts (Xu et al., 2024) —when internal parametric knowledge conflicts with external contextual knowledge, LLMs may favor internal parametric knowledge, leading to contextual faithfulness hallucinations (Bi et al., 2024; Ming et al., 2025; Niu et al., 2024). Such hallucinations are particularly critical in knowledge-intensive domains (Vishwanath et al., 2024) like rare disease medical consultations (Reese et al., 2025), where clinicians may lack systematic knowledge reserves (Zhang et al., 2022) to judge whether model responses are faithful to contexts, while patient communities often rely on self-consultation or LLM queries without professional medical supervision (Busch et al., 2025; Aydin et al., 2025). Chen & Shu (2024); Zhang et al. (2025c) shows LLM-generated content is more deceptive than human-written content. Without clear attributability, faithfulness hallucinations pose potential risks to clinical decisions and patient behaviors (Kim et al., 2025).

Current research primarily follows two directions in enhancing the reliability of LLMs: (i) generation with citations, where models produce responses accompanied by attributable citations (Wu

et al., 2025; Abolghasemi et al., 2025; Ji et al., 2025; Press et al., 2024; Song et al., 2025), and (ii) improving contextual faithfulness through techniques such as prompting strategies (Zhou et al., 2023; Zhang et al., 2025a), constrained decoding (Shi et al., 2024; T.y.s.s et al., 2025; Liu et al., 2025), or fine-tuning (Bi et al., 2025; Huang et al., 2025b; Si et al., 2025; Li et al., 2025a). However, the former struggles to ensure consistency between the generated content and its cited sources, while the latter typically lacks mechanisms for explicit attribution. Consequently, achieving both faithfulness and verifiable attribution remains a critical and unresolved challenge.

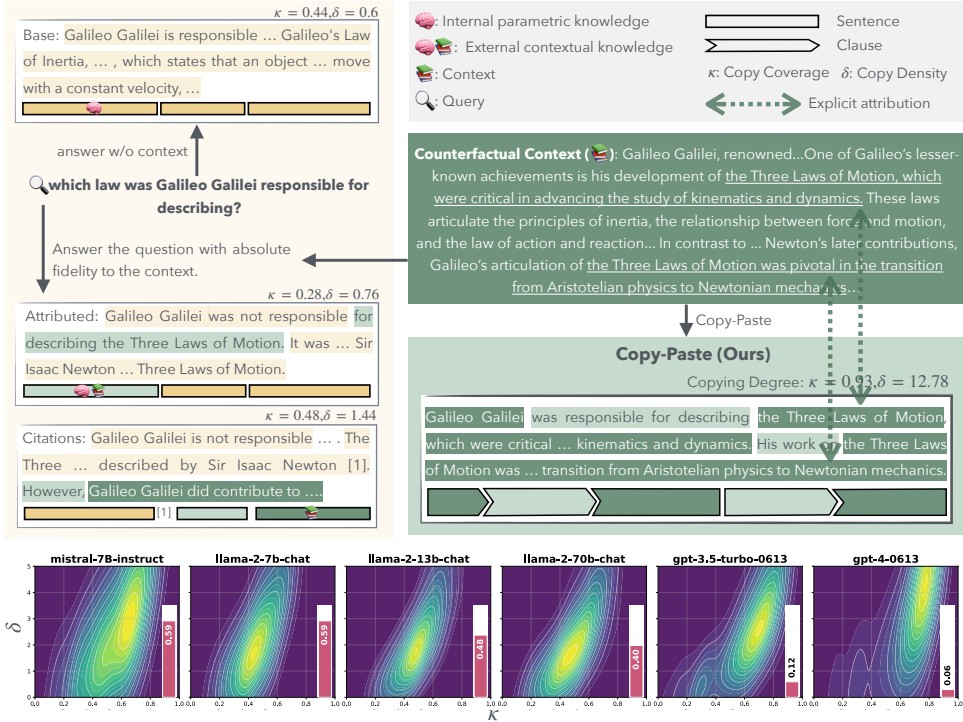

Figure 1: Upper: Response composition patterns comparison between Copy-Paste and mainstream approaches. Lower: Inverse correlation between copying degree and faithfulness hallucination across different models. Kernel ■ show copying degree; Bar ■ show hallucination.

To address these challenges, we propose an intuitive solution: rather than having models reinterpret retrieved content, we advocate for directly quoting original sentences. This copy-paste generation strategy embeds key contextual fragments directly, avoiding secondary knowledge processing and potentially reducing paraphrasing hallucination risks. Importantly, copied content itself serves as direct evidence of faithfulness without requiring additional verifiable attribution mechanism. This approach is motivated by our observation of an inverse correlation between copying degree and hallucination density on the RAGTruth dataset (Figure 1), leading us to hypothesize that high copying degrees may help mitigate hallucination problems.

Specifically, we formally propose Copy-Paste as a generation paradigm that leverages high-copying degree as an operational proxy for contextual faithfulness through a two-stage pipeline that internalizes surface-level copying behavior into model-level contextual trust. The first stage generates high-copying responses through hard and soft constraints to enhance copying degree. The second stage (**CopyPasteLLM**) applies direct preference optimization (Rafailov et al., 2023) training to internalize the high-copying preferences from the first stage into the LLM's contextual faithfulness. Experimental results demonstrate that CopyPasteLLM, trained on only 365 high-copying samples, outperforms strongest baselines by 12.2%-24.5% on FaithEval. Additionally, we propose the **Context-Parameter Copying Capturing** algorithm, which enables fine-grained analysis of knowledge source reliance throughout the entire Chain-of-Thought reasoning process, rather than merely examining final short answers. The algorithm captures contextual versus parametric knowledge usage at each token position, providing novel insights into how models dynamically balance different knowledge sources during sequential reasoning. Mechanistic analysis reveals CopyPasteLLM

maintains similar contextual knowledge representations as the base model while recalibrating internal confidence in parametric knowledge, thereby enhancing contextual trust.

## 2 PRELIMINARIES

### 2.1 PROBLEM FORMULATION

**Task**   Given a query $Q$ and a context $C$, the model generates an answer $A$. In high-stakes domains such as medicine, the faithfulness of the generated answer to the context is of paramount importance. While conventional RAG research often emphasizes abstractive generation and semantic relevance, our focus in this work is a specialized task that we term **Copy-Paste**. The goal of Copy-Paste is to maximize the reuse of lexical units from the context $C$ in the final answer $A$, thereby ensuring high contextual faithfulness and minimizing hallucination. Formally, the task can be defined as: $(Q, C) \mapsto A$.

**Quantification**   Following Grusky et al. (2018), we quantify the response copying degree from context with two metrics:

$$\kappa = \frac{1}{|A|} \sum_{f \in \mathcal{F}} |f|, \quad \delta = \frac{1}{|A|} \sum_{f \in \mathcal{F}} |f|^2 \tag{1}$$

where $\mathcal{F}$ is the set of copy fragments computed by copy fragment detection algorithm (detailed at Appendix I), $|\cdot|$ denotes sequence length. **Copy Coverage** ($\kappa$): the fraction of answer tokens that are covered by some copy fragment, reflecting the overall degree of lexical reuse. **Copy Density** ($\delta$): a length-sensitive variant that emphasizes longer copied fragments, capturing whether the answer tends to copy long spans verbatim rather than isolated words.

**Balance**   While maximizing copy-paste is central to our formulation, an effective answer $A$ should also remain relevant to the query $Q$ and be linguistically fluent. Specifically, we measure query relevance using embedding-based similarity, and fluency via perplexity. Thus, the Copy-Paste task can be viewed as optimizing a trade-off among **faithfulness**, **query relevance**, and **fluency**. Unlike extractive summarization (Zhang et al., 2023), Copy-Paste is query-aware and ensures fluent, context-faithful answers.

### 2.2 MOTIVATING OBSERVATION ON RAGTRUTH

To validate the intuition that high copying degrees may reduce hallucination, we conducted a preliminary analysis on the RAGTruth QA subset Niu et al. (2024), which contains 839 context-dependent questions. Each question includes responses from 6 different models with word-level contextual faithfulness hallucination annotations, enabling precise quantification of hallucination density per model.

We computed copy coverage ($\kappa$) and copy density ($\delta$) for each model's responses across the dataset, then visualized the relationship using two-dimensional kernel density estimation with copy coverage (x-axis) and copy density (y-axis). The analysis reveals a clear pattern: density kernels positioned toward the upper-right region (indicating higher copying coverage and density) correspond to lower hallucination density across models (Figure 1).

## 3 METHODOLOGY

Our approach consists of two sequential stages: (1) constructing high-copying candidate responses through Copy-Paste-Prompting methods, and (2) training CopyPasteLLM through automated preference data construction that internalizes a preference for contextual evidence. Figure 2 illustrates the complete pipeline. To verify that the learned policy truly reallocates reliance from parametric priors to context, we additionally introduce an interpretability tool, Context-Parameter Copying Capturing.

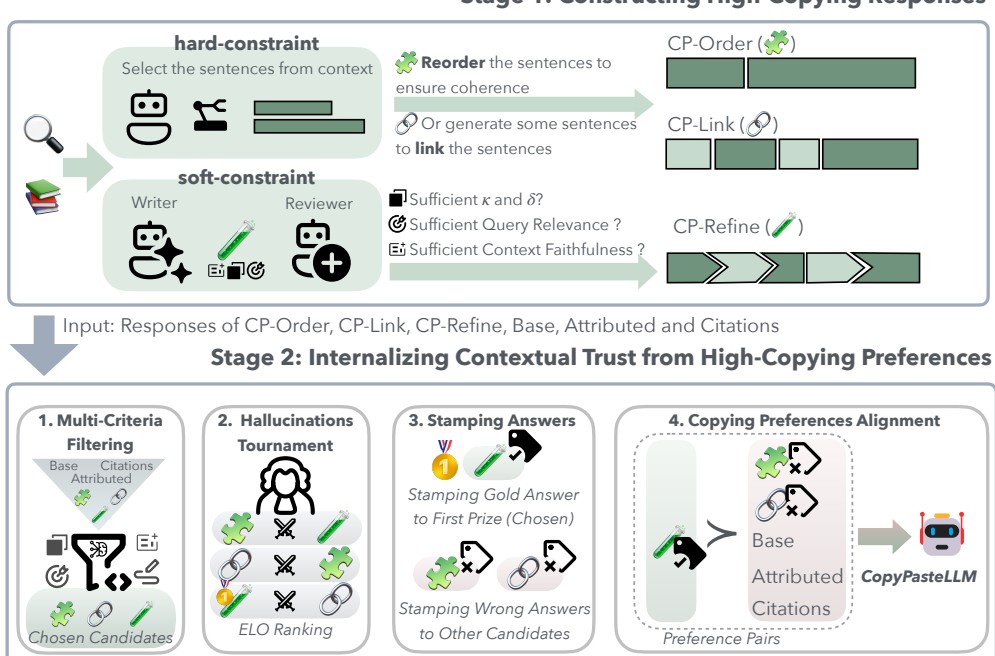

Figure 2: Two-stage Copy-Paste pipeline: Stage 1 constructs high-copying responses; Stage 2 filters, judges, stamps answers, and aligns preferences to train CopyPasteLLM.

### 3.1 COPY-PASTE-PROMPTING: CONSTRUCTING HIGH-COPYING RESPONSES

We operationalize the Copy-Paste objective through three complementary prompting paradigms that progressively relax constraints while preserving lexical fidelity to the context. CP-Order implements a strict extractive regime: it first selects context sentences relevant to the query and then directly reorders them into a coherent answer. This hard constraint intentionally forgoes abstractive para-phrasing, which suppresses the model's tendency to resolve conflicts using parametric priors. The method excels when answers can be composed from a small set of highly informative sentences but tends to sacrifice fluency when discourse connectives are missing. (See L.1.1 & L.1.2 for prompts)

CP-Link maintains the same extractive core but allows the model to generate short transitions be-tween copied spans. These transitions are not intended to introduce new facts; instead, they serve as discourse glue to restore local coherence after sentence reordering. Empirically, this limited gen-erative freedom improves readability while preserving the high-copying signature that anchors the answer to source text. (See L.1.1 & L.1.3 for prompts)

In contrast, CP-Refine adopts a soft-constraint, iterative refinement process with a writer–reviewer loop. The writer proposes an answer given the query and context; the reviewer provides verbal feed-back focused on copying degree, contextual faithfulness, query relevance, and fluency; the writer then revises the answer until a composite copy score exceeds a threshold. This procedure treats copying as a target state that is continually optimized rather than a fixed structural constraint. As shown by our experiments (See Table 2), CP-Refine achieves a better balance among faithfulness, readability, and relevance (See L.1.4 for prompts). Algorithm 1 in Appendix summarizes the unified procedure, which we use to produce diverse yet consistently high-copying candidates for down-stream preference construction.

### 3.2 COPYPASTELLM: INTERNALIZING CONTEXTUAL TRUST FROM HIGH-COPYING PREFERENCES

Copy-Paste-Prompting supplies not only single responses but a structured spectrum of behav-iors—from strictly extractive to softly refined. CopyPasteLLM converts this spectrum into explicit

preferences that can be internalized by a policy through direct preference optimization. Our pipeline begins by generating six types of candidates for each query–context pair: conventional abstractive baselines (Base, Attributed, Citations) and three Copy-Paste variants (CP-Order, CP-Link, CP-Refine). We then perform multi-criteria filtering that simultaneously enforces contextual faithfulness (AlignScore, MiniCheck), copying strength ($\kappa$, $\delta$), query relevance (embedding similarity), and fluency (perplexity). This step ensures the retained set covers a high-quality front of the faithfulness–fluency–relevance trade space rather than merely maximizing copying.

The remaining candidates are ranked by an Elo-style LLM-as-Judge tournament that diagnoses two major hallucination modes—Twist and Causal—so the final preference reflects error severity, not only stylistic quality. A key nuance arises when gold answers are available: we append the correct answer to the top Copy-Paste candidate to transform faithful reasoning into a definitive conclusion, while appending incorrect answers to the other Copy-Paste candidates to create informative negative pairs. This labeling strategy focuses learning on trusting context while disentangling reasoning traces from final decisions. The resulting dataset yields roughly five preference pairs per sample, enabling data-efficient DPO training that teaches the model to prefer high-copying, context-grounded responses even when they conflict with parametric priors. Algorithm 2 in Appendix formalizes the procedure.

### 3.3 CONTEXT-PARAMETER COPYING CAPTURING

Context-Parameter Copying Capturing provides a principled, token-level probe of knowledge usage during generation. The method executes two runs for each query: with context and without context. At each decoding step in Chain-of-Thought mode, it collects the top-$K$ candidate tokens with their probabilities and hidden states. Tokens that appear in the provided context are taken as contextual knowledge, whereas tokens that are preferred in the context-free run serve as proxies for parametric knowledge. Algorithm 4 specifies the full procedure.

Conceptually, this procedure is inspired by Knowledge Token Capturing (KTC) (Bi et al., 2024). Unlike KTC, which primarily analyzes short final answers, our Context-Parameter Copying Capturing extends the analysis to the entire Chain-of-Thought response trajectory, enabling sequential, position-aware assessment of contextual versus parametric reliance.

## 4 EXPERIMENT

Our Copy-Paste approach is a two-stage framework where Copy-Paste-Prompting generates high-copying preference data, and CopyPasteLLM learns contextual faithfulness from this data. To validate our complete pipeline, we conduct comprehensive experiments addressing three key research questions:

- **RQ1**: Do Copy-Paste-Prompting methods effectively enhance contextual faithfulness and mitigate RAG hallucinations through high-copying response generation?

- **RQ2**: Does training with high-copying responses from Copy-Paste-Prompting as DPO preference trajectories enable CopyPasteLLM to genuinely trust contextual knowledge—even when it is counterfactual?

- **RQ3**: What are the underlying mechanisms of CopyPasteLLM's contextual belief? We will interpret this by analyzing logits and hidden states.

### 4.1 TWO-STAGE FRAMEWORK VALIDATION

Experimental setup is detailed in Appendix B.

#### 4.1.1 STAGE 1: COPY-PASTE-PROMPTING AS PREFERENCE DATA GENERATOR (RQ1)

In the first stage, we evaluate whether our prompting methods can effectively generate responses with high-copying and improved contextual faithfulness. The baselines here represent different response generation paradigms that will serve as rejected responses in our CopyPasteLLM training. Our primary objectives are to: (1) validate that Copy-Paste-Prompting methods achieve superior

Table 1: Counterfactual scenarios: Performance comparison of CopyPasteLLM against baselines. We removed 241 samples used for training CopyPasteLLM from FaithEval, with the remaining samples used for testing (detailed in the RQ2 setup of Appendix Table 4). Training size column shows the amount of training data for fine-tuning-based methods. [T] indicates seen data for the respective model. **Bold** values highlight the best performing method in unseen settings.

| Model | Method | Training Size | FaithEval Acc | FaithEval Hit | ConFiQA-QA Acc | ConFiQA-QA Hit | ConFiQA-MR Acc | ConFiQA-MR Hit | ConFiQA-MC Acc | ConFiQA-MC Hit |
|---|---|---|---|---|---|---|---|---|---|---|
| Llama-3-8B | Context-DPO (Bi et al., 2025) | 18,000 | 80.2 | 36.7 | 88.9[T] | 96.1[T] | 88.4[T] | 85.8[T] | 92.1[T] | 80.9[T] |
| | Attributed (Zhou et al., 2023) | - | 67.1 | 34.2 | 51.5 | 91.4 | 53.3 | 71.5 | 37.3 | 53.6 |
| | CoCoLex (T.y.s.s et al., 2025) | - | 69.2 | 17.9 | 48.5 | 37.4 | 53.9 | 14.8 | 36.1 | 15.5 |
| | Canoe (Si et al., 2025) | 10,000 | 71.4 | 34.0 | 64.3 | 93.2 | 66.6 | **83.8** | 64.5 | 73.7 |
| | ParamMute (Huang et al., 2025b) | 32,580 | 68.5 | 22.5 | 74.4 | 82.2 | 75.5 | 72.4 | 81.4 | 70.2 |
| | CopyPasteLLM (Ours) | **365** | **92.8** | **37.2** | **83.6** | **96.7** | **80.9** | 83.4 | **86.8** | **75.9** |
| Mistral-7B-v0.2 | Context-DPO (Bi et al., 2025) | 18,000 | 77.1 | 33.8 | 84.8[T] | 94.8[T] | 81.3[T] | 85.3[T] | 80.4[T] | 80.8[T] |
| | Attributed (Zhou et al., 2023) | - | 65.6 | 32.0 | 56.6 | 84.4 | 29.2 | 69.8 | 39.0 | 57.4 |
| | CoCoLex (T.y.s.s et al., 2025) | - | 65.3 | 35.4 | 57.3 | 50.8 | 41.8 | 33.5 | 32.5 | 33.7 |
| | CopyPasteLLM (Ours) | **365** | **89.3** | **41.8** | **84.4** | **95.0** | **80.8** | **90.8** | **82.5** | **86.3** |
| Llama-3.1-8B | Attributed (Zhou et al., 2023) | - | 65.5 | 32.0 | 49.9 | 88.4 | 39.8 | 69.2 | 15.5 | 52.6 |
| | CoCoLex (T.y.s.s et al., 2025) | - | 68.1 | 36.2 | 48.5 | 57.3 | 40.4 | 38.4 | 13.5 | 37.2 |
| | CopyPasteLLM (Ours) | 365 | **92.6** | **41.0** | **72.4** | **90.1** | **75.4** | **84.8** | **83.5** | **79.9** |

contextual faithfulness through explicit copying mechanisms, and (2) generate high-quality preferred responses for subsequent DPO training. A comprehensive comparison with state-of-the-art methods will be presented in the next stage after DPO training.

Table 2: Performance comparison of Copy-Paste-Prompting against baselines across models and datasets. Methods with colored backgrounds are our proposed Copy-Paste-Prompting. **Bold** indicates the best performance, underlined indicates the second-best performance. *Faith.*: Faithfulness (*M.C.*: MiniCheck, *A.S.*: AlignScore), *Hallu.*: Hallucination, *Flu.*: Fluency.

| Method | RAGTruth Faith. M.C. | Faith. A.S. | Hallu. Twist | Hallu. Causal | Flu. | FaithEval Faith. M.C. | Faith. A.S. | Hallu. Twist | Hallu. Causal | Flu. | PubmedQA Faith. M.C. | Faith. A.S. | Hallu. Twist | Hallu. Causal | Flu. | AVERAGE Faith. | Hallu. | Flu. |
|---|---|---|---|---|---|---|---|---|---|---|---|---|---|---|---|---|---|---|
| *Mistral-7B-Instruct-v0.2 (7B)* | | | | | | | | | | | | | | | | | | |
| Attributed | 69.58 | 63.43 | 1506.9 | 1494.5 | 19.54 | 88.28 | 90.67 | 1527.1 | 1513.7 | 37.32 | 75.49 | 77.90 | 1464.7 | 1450.4 | 23.53 | 77.56 | 1492.9 | 26.80 |
| Citations | 57.82 | 49.39 | 1472.5 | 1475.7 | 14.41 | 73.50 | 74.25 | 1392.1 | 1416.2 | 27.98 | 55.79 | 52.35 | 1415.9 | 1370.0 | 13.93 | 60.52 | 1423.7 | 18.77 |
| CP-Link | 89.39 | 75.45 | 1518.9 | 1519.5 | 73.33 | 93.41 | 92.44 | 1510.9 | 1521.9 | 49.40 | 96.50 | 88.52 | 1518.4 | 1580.7 | 35.57 | 89.29 | 1528.4 | 52.77 |
| CP-Order | 91.25 | 71.98 | 1467.9 | 1472.4 | 65.62 | 94.89 | 92.27 | 1522.6 | 1501.5 | 43.74 | 93.18 | 82.35 | 1528.3 | 1559.1 | 32.65 | 87.65 | 1508.6 | 47.34 |
| CP-Refine | 82.18 | 74.56 | 1533.8 | 1537.9 | 18.46 | 92.85 | 94.68 | 1547.4 | 1546.7 | 26.63 | 91.52 | 88.21 | 1572.7 | 1539.7 | 17.79 | 87.33 | 1546.4 | 20.96 |
| *Llama-3.1-8B-Instruct (8B)* | | | | | | | | | | | | | | | | | | |
| Attributed | 57.02 | 65.29 | 1526.3 | 1554.3 | 26.22 | 85.22 | 85.65 | 1516.5 | 1536.9 | 330.8 | 71.10 | 60.01 | 1530.0 | 1553.1 | 47.36 | 70.72 | 1536.2 | 134.8 |
| Citations | 64.27 | 72.81 | 1428.5 | 1574.4 | 16.78 | 88.81 | 86.80 | 1486.2 | 1555.6 | 39.65 | 78.56 | 73.03 | 1403.4 | 1463.4 | 19.11 | 77.38 | 1485.3 | 25.18 |
| CP-Link | 70.58 | 78.83 | 1401.1 | 1328.3 | 17.83 | 91.54 | 89.23 | 1456.2 | 1366.3 | 24.09 | 80.74 | 80.79 | 1396.4 | 1371.1 | 19.65 | 81.95 | 1386.6 | 20.52 |
| CP-Order | 75.30 | 94.81 | 1498.4 | 1498.0 | 26.35 | 95.44 | 98.12 | 1523.2 | 1541.2 | 33.46 | 87.07 | 97.62 | 1633.6 | 1559.1 | 27.83 | 91.39 | 1542.3 | 29.21 |
| CP-Refine | 77.30 | 88.52 | 1645.7 | 1545.0 | 17.75 | 94.40 | 93.71 | 1517.9 | 1500.1 | 26.99 | 87.29 | 91.19 | 1536.5 | 1553.2 | 18.64 | 88.74 | 1549.7 | 21.13 |
| *Qwen2.5-72B-Instruct (72B)* | | | | | | | | | | | | | | | | | | |
| Attributed | 57.00 | 62.23 | 1504.5 | 1525.5 | 19.68 | 85.74 | 83.03 | 1537.3 | 1490.0 | 293.8 | 77.99 | 69.25 | 1509.9 | 1441.5 | 33.42 | 72.54 | 1501.5 | 115.6 |
| Citations | 74.32 | 77.52 | 1455.5 | 1498.0 | 18.61 | 90.98 | 88.30 | 1456.5 | 1476.7 | 34.67 | 82.01 | 76.62 | 1358.8 | 1413.6 | 22.89 | 81.63 | 1443.2 | 25.39 |
| CP-Link | 75.75 | 85.37 | 1446.3 | 1363.2 | 27.47 | 92.88 | 92.00 | 1443.5 | 1424.2 | 39.55 | 86.21 | 88.58 | 1527.9 | 1489.2 | 33.43 | 86.80 | 1449.1 | 33.48 |
| CP-Order | 76.32 | 94.60 | 1509.2 | 1589.6 | 30.56 | 95.78 | 98.16 | 1539.3 | 1579.7 | 38.11 | 87.85 | 97.52 | 1546.8 | 1575.9 | 35.26 | 91.71 | 1556.8 | 34.65 |
| CP-Refine | 78.14 | 90.88 | 1584.6 | 1523.7 | 20.12 | 94.72 | 95.48 | 1523.4 | 1529.4 | 27.65 | 88.88 | 95.04 | 1556.7 | 1579.9 | 20.29 | 90.52 | 1549.6 | 22.69 |
| *DeepSeek-V3-0324 (671B)* | | | | | | | | | | | | | | | | | | |
| Attributed | 56.42 | 59.60 | 1417.1 | 1449.1 | 27.52 | 86.90 | 83.46 | 1524.3 | 1535.0 | 63.27 | 75.56 | 69.24 | 1449.2 | 1487.9 | 36.88 | 71.86 | 1477.1 | 42.56 |
| Citations | 62.32 | 64.45 | 1510.8 | 1565.6 | 34.63 | 87.38 | 85.69 | 1463.0 | 1477.0 | 36.09 | 75.93 | 71.85 | 1460.4 | 1387.5 | 23.27 | 74.60 | 1477.4 | 31.33 |
| CP-Link | 70.59 | 72.54 | 1382.9 | 1360.3 | 34.19 | 92.60 | 88.08 | 1489.1 | 1374.8 | 35.55 | 81.56 | 77.67 | 1380.9 | 1351.1 | 28.54 | 80.51 | 1389.9 | 32.76 |
| CP-Order | 75.53 | 92.87 | 1579.4 | 1555.2 | 59.11 | 95.23 | 97.79 | 1569.9 | 1548.1 | 34.30 | 87.20 | 97.38 | 1561.8 | 1621.7 | 27.56 | 91.00 | 1572.7 | 40.32 |
| CP-Refine | 77.14 | 90.02 | 1609.8 | 1569.7 | 22.57 | 94.45 | 93.06 | 1453.7 | 1565.2 | 33.84 | 87.39 | 91.05 | 1647.7 | 1651.7 | 21.91 | 88.85 | 1583.0 | 26.11 |

Our experimental results demonstrate that Copy-Paste-Prompting methods consistently outperform baselines across all evaluation metrics (Table 2). **(1) CP-Refine** excels in hallucination reduction (best in 3/4 models, 14/24 top scores) and contextual faithfulness (+10.9% to 19.1% over baselines) while maintaining fluency—achieving best perplexity in Q-72B/D-V3 and second-best in M-7B/L-8B, suggesting advanced models better handle high-copying constraints. **(2) CP-Order** leads contextual faithfulness (14/24 top scores) with second-best hallucination performance but notably poorer fluency. **(3) CP-Link** shows modest improvements, excelling only in contextual faithfulness

with even worse fluency than CP-Order, indicating hard constraints limit generative capabilities. **(4)** We observe **strong hallucination-faithfulness correlation**: in 18/24 scenarios (75%), optimal hallucination performance coincides with best contextual faithfulness. We hypothesize that the superior contextual faithfulness of Copy-Paste-Prompting stems from high-copying in responses. Copy-Paste-Prompting achieves significantly higher copying degree than the two baselines (see Appendix Figure 5). Additionally, we compare query relevance between the three Copy-Paste-Prompting methods and the strongest baseline in Appendix Figure 6, demonstrating that Copy-Paste-Refine can address queries while maintaining high copying rates through soft constraints.

### 4.1.2 STAGE 2: COPYPASTELLM (RQ2)

Table 3: Accuracy in non-counterfactual settings. PubMedQA is evaluated on artificial subset 20,000 samples (none used for CopyPasteLLM training, see Appendix Table 4). ConFiQA uses Original context and Original answers.

| Method | Mistral-7B-v0.2 | | | | Llama-3-8B | | | | Llama-3.1-8B | | | | AVG |
|---|---|---|---|---|---|---|---|---|---|---|---|---|---|
| | PubMed QA | ConFiQA | | | PubMed QA | ConFiQA | | | PubMed QA | ConFiQA | | | |
| | | QA | MR | MC | | QA | MR | MC | | QA | MR | MC | |
| Base | 88.60 | 96.22 | 71.20 | 72.27 | 97.3 | 98.02 | 93.00 | 91.02 | **98.15** | 97.93 | 89.48 | 89.97 | 90.26 |
| CopyPasteLLM (Ours) | **91.40** | **97.43** | **91.87** | **91.20** | **97.5** | **99.30** | **97.17** | **96.27** | 97.67 | **99.02** | **94.95** | **94.92** | **95.73** |

CopyPasteLLM demonstrates remarkable efficiency by achieving superior performance in counterfactual scenarios using only 365 query-context pairs as input to construct preference data through our automated pipeline—a base data requirement that is $50\times$ smaller than the strongest baseline Context-DPO (18,000 samples) and significantly more efficient than other fine-tuning methods such as Canoe (10,000) and ParamMute (32,580). As shown in Table 1, on the FaithEval counterfactual subset, CopyPasteLLM surpasses the strongest baselines by substantial margins: 12.6, 12.2, and 24.5 percentage points across Llama-3-8B, Mistral-7B-v0.2, and Llama-3.1-8B respectively, achieving a peak accuracy of 92.8% on Llama-3-8B—remarkably outperforming GPT-4o's reported 47.5% on this challenging subset (see Appendix Table 6). Additionally, CopyPasteLLM consistently achieves the highest Hit Rate across all models, despite the inherent difficulty of exact matching in FaithEval's lengthy gold standard answers. On ConFiQA's three counterfactual subsets, CopyPasteLLM maintains superior performance in unseen settings compared to recent fine-tuning baselines and copy-guided decoding method CoCoLex, with particularly notable results on Mistral-7B-v0.2 where it outperforms even Context-DPO trained on ConFiQA on the most challenging Multi-Conflict subset.

In non-counterfactual scenarios, CopyPasteLLM maintains exceptional contextual faithfulness while demonstrating significant improvements over base models (Table 3). On relatively straightforward datasets—PubMedQA and ConFiQA-QA—the method achieves modest but consistent improvements, with average accuracy gains of 1.01% (from 96.04% to 97.05%). More importantly, on the more challenging ConFiQA-MR and ConFiQA-MC subsets, CopyPasteLLM delivers substantial performance gains, improving average accuracy from 84.49% to 94.37%, with the most dramatic improvement of 20.67% observed on Mistral-7B-v0.2 for the MR subset. These results demonstrate that CopyPasteLLM's enhanced contextual trust, achieved without introducing additional parametric knowledge through LoRA training, leads to significant improvements in knowledge-intensive question answering accuracy. For fine-grained analysis across conflict complexity, knowledge domains, and reasoning ambiguity, see Appendix E; for response length and copying behavior analysis, see Appendix F; for ablation studies and training dynamics, see Appendix G.

### 4.2 INTERPRETABLE ANALYSIS OF COPYPASTELLM (RQ3)

We propose the Context-Parameter Copying Capturing (Algorithm 4), which is designed to capture the degree to which the model copies contextual or parametric knowledge during token generation. Specifically, in CoT reasoning mode, our method monitors the model's internal representations by analyzing the top-K token logits (ranked by probability) and corresponding hidden states at each generation step, thereby quantifying the model's reliance on external context versus internal parametric knowledge. This algorithm extends the Knowledge Token Capturing (Bi et al., 2024) to sequential analysis, enabling comprehensive evaluation of model responses during CoT reasoning.

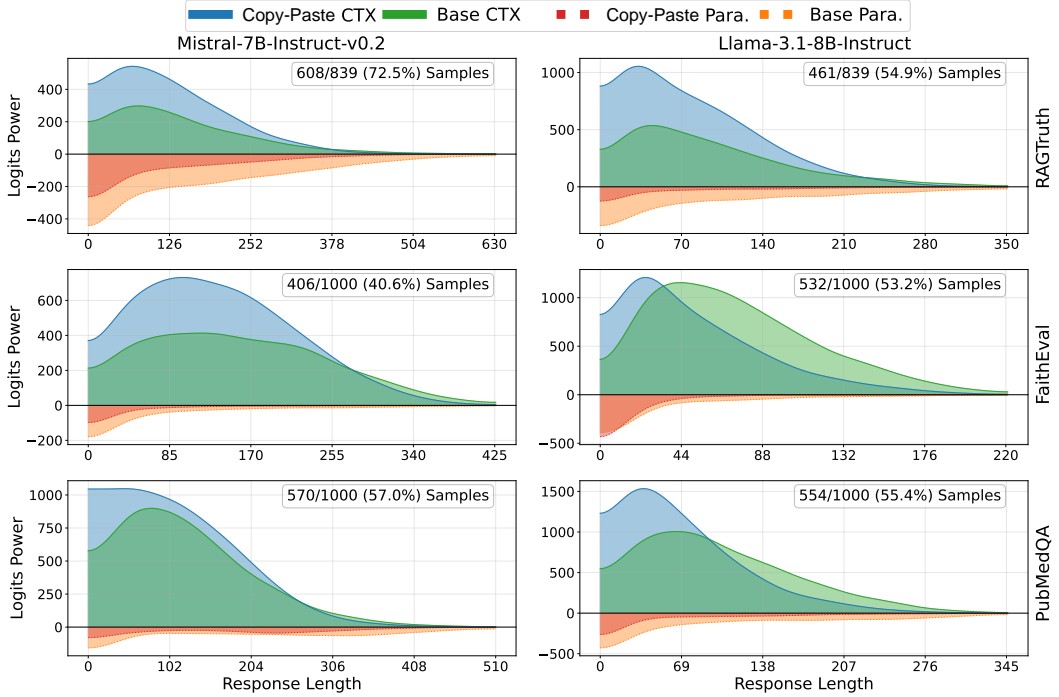

Figure 3: Logits power distribution across response lengths for contextual (CTX) and parametric (Para.) knowledge. Values above x=0 indicate CTX logits power, values below x=0 indicate Para. logits power (negated for visualization).

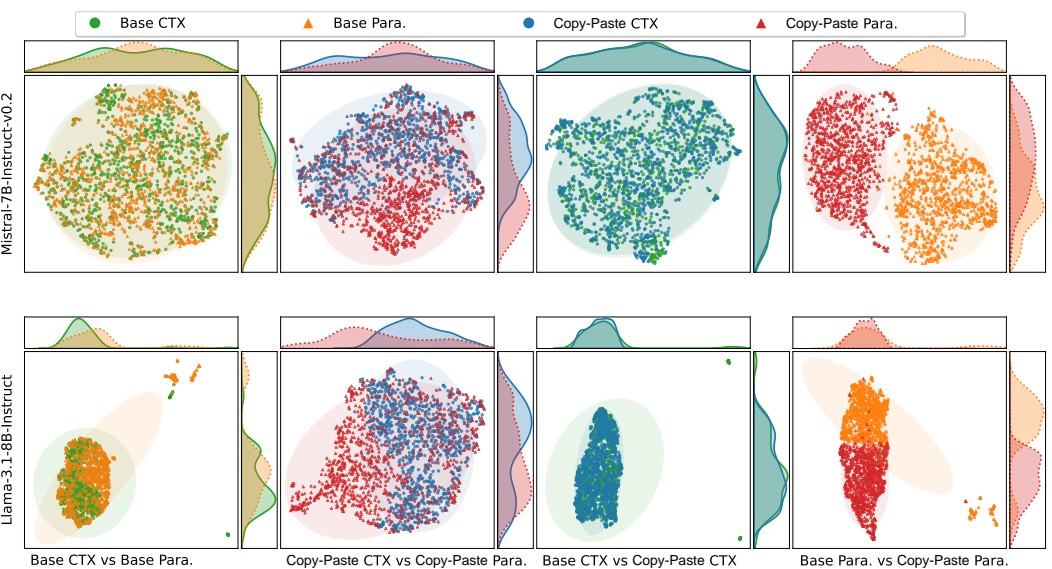

Figure 4: Dimensionality reduction visualization of hidden states distributions between contextual (CTX) and parametric (Para.) knowledge on PubMedQA dataset across two base models. Each subplot shows pairwise comparisons with marginal KDE distributions and confidence ellipses. See Appendix Figures 15 and 14 for RAGTruth and FaithEval.

We first analyze the logits output power of CopyPasteLLM and its base models across three datasets at each generation step, considering both the magnitude and frequency of logits at specific response positions, as illustrated in Figure 3. To ensure fair comparison by providing base with longer token generation opportunities, we filtered out samples where CopyPasteLLM responses exceeded base response lengths, with complete dataset statistics shown in Appendix Figure 13. Our analysis reveals three key observations: (1) In CoT with context task, Both base and CopyPasteLLM demonstrate higher reliance on contextual knowledge than parametric knowledge. (2) However, CopyPasteLLM exhibits significantly stronger contextual knowledge utilization compared to base, while showing reduced reliance on parametric knowledge. (3) From a positional perspective, CopyPasteLLM achieves peak contextual knowledge utilization earlier in the response generation process than base. Collectively, these findings suggest that CopyPasteLLM not only demonstrates stronger but also earlier contextual engagement compared to base, indicating enhanced contextual trust and willingness to *believe* the provided context.

We further employ UMAP dimensionality reduction to analyze the captured hidden states distributions, as shown in Figure 4. Our visualization reveals two striking patterns: (1) Base models exhibit minimal distinction between contextual and parametric knowledge semantic representations (1st column), whereas CopyPasteLLM demonstrates relatively clear separation between these two knowledge types (2nd column). (2) More intriguingly, contextual knowledge representations in CopyPasteLLM remain nearly co-distributed with those in base models (3rd column), while their parametric knowledge distributions differ substantially (4th column). Based on these observations, we infer that CopyPasteLLM fundamentally recalibrates the model's internal confidence in parametric knowledge without compromising its contextual processing capabilities. This selective parametric knowledge suppression, rather than contextual knowledge enhancement, enables CopyPasteLLM to achieve superior contextual faithfulness by strategically reducing competition from internal parametric knowledge during generation. For a theoretical interpretation of this mechanism through attention dynamics and entropy reduction, see Appendix A.

## 5 RELATED WORK

While Retrieval-Augmented Generation (RAG) has emerged as a promising paradigm for grounding large language models in external knowledge (Fan et al., 2024; Zhao et al., 2024), ensuring contextual faithfulness remains an open challenge. LLMs often exhibit a tendency to rely on their pretrained parametric knowledge rather than adhering to the provided context, resulting in responses that may contradict or ignore retrieved evidence (Niu et al., 2024; Bi et al., 2024; Ming et al., 2025). This contextual unfaithfulness poses significant concerns in critical applications such as healthcare (Vishwanath et al., 2024; Kim et al., 2025), where accuracy and reliability are paramount.

Existing research has systematically studied this phenomenon from evaluation and mechanistic perspectives. Evaluation studies construct synthetic scenarios revealing LLMs' propensity to favor internal knowledge over external evidence (Xu et al., 2024; Li et al., 2025b; Joren et al., 2025; Goyal et al., 2025). Mechanistic analyses identify attention heads (Wu et al., 2024; Huang et al., 2025a), FFNs (Sun et al., 2024) and logit distributions (Bi et al., 2024) that respectively process external and internal knowledge sources.

Solutions to improve contextual faithfulness include generation with citations (Gao et al., 2023; Press et al., 2024; Song et al., 2025; Wu et al., 2025), prompt engineering (Zhou et al., 2023; Zhang et al., 2025a), decoding methods (Shi et al., 2024; T.y.s.s et al., 2025; Liu et al., 2025) and finetuning (Bi et al., 2025; Si et al., 2025; Li et al., 2025a; Huang et al., 2025b). While generation with citations methods may lack content-source consistency and other approaches often provide limited attribution mechanisms, our Copy-Paste paradigm targets both challenges simultaneously: it enhances contextual faithfulness through direct lexical reuse from source text while inherently providing transparent attribution, and internalizes this copying behavior into genuine model-level contextual trust through preference optimization.

## 6 CONCLUSION

We propose Copy-Paste, a generation paradigm that directly embeds contextual fragments into responses to mitigate faithfulness hallucinations in RAG systems. Based on the observed inverse

correlation between copying degree and hallucination density, we instantiate this paradigm through a two-stage framework: Copy-Paste-Prompting methods first generate high-copying responses, then preference optimization internalizes contextual trust into CopyPasteLLM. CopyPasteLLM achieves remarkable data efficiency, delivering 12.2%-24.5% improvements on FaithEval using only 365 training samples—50× smaller than existing baselines. Our Context-Parameter Copying Capturing analysis reveals that effectiveness stems from recalibrating parametric knowledge confidence rather than enhancing contextual representations. The copy-paste paradigm provides an elegant solution to RAG attribution challenges, where copied content serves as inherent faithfulness evidence without requiring additional verification mechanisms. We discuss limitations and future directions in Appendix K.

## 7 ETHICS STATEMENT

This work addresses the critical challenge of contextual faithfulness in large language models, particularly in high-stakes domains such as healthcare. While our CopyPasteLLM approach aims to reduce hallucinations by promoting direct copying from provided context, we acknowledge potential risks: over-reliance on copied content may lead to verbatim reproduction of potentially biased or incorrect source material. The method's effectiveness depends on the quality and accuracy of the provided context, and users should exercise caution when applying this approach in sensitive applications. We encourage responsible deployment with appropriate human oversight and validation mechanisms.

## 8 REPRODUCIBILITY STATEMENT

To ensure reproducibility, we provide the following: (1) All experimental details and hyperparameters are documented in the appendix. (2) We use publicly available datasets (FaithEval, ConFiQA, PubMedQA, RAGTruth) with standard evaluation protocols (see Appendix B). (3) Model training details, including DPO hyperparameters (see Appendix D) and preference data construction procedures (see Algorithm 1 and 2). (4) The Context-Parameter Copying Capturing algorithm is fully described in Algorithm 4. (5) All prompting templates for Copy-Paste-Prompting methods are provided in Appendix L. The complete implementation is available at `https://github.com/longyongchao/CopyPasteLLM`.

## ACKNOWLEDGMENTS

We sincerely thank the anonymous reviewers and area chairs for their insightful comments and constructive suggestions that helped improve this paper. This work was supported by the National Natural Science Foundation of China (62102008, 62172018, 62202332, 62376197, 62020106004, 92048301), the CCF-Tencent Rhino-Bird Open Research Fund (CCF-Tencent RAGR20250108), the CCF-Zhipu Large Model Innovation Fund (CCF-Zhipu202414), the Tianjin Science and Technology Program (23JCYBJC00360), the Key Research and Development Program of Shaanxi Province (2023-ZDLGY-48), the Tianchi Elite Youth Doctoral Program (CZ002701, CZ002707), the PKU-OPPO Fund (BO202301, BO202503), the Research Project of Peking University in the State Key Laboratory of Vascular Homeostasis and Remodeling (2025-SKLVHR-YCTS-02), and the Beijing Municipal Science and Technology Commission (Z251100000725008).

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

## A  MECHANISTIC INTERPRETATION OF COPY-PASTE EFFECTIVENESS

In this section, we provide a mechanistic interpretation explaining how the external constraint of high-copying responses translates into improved contextual faithfulness and reduced hallucinations within the LLM's internal dynamics. This interpretation connects the Copy-Paste objective to fundamental model components, which we analyze from two perspectives: (1) Attention Dynamics and Contextual Anchoring, and (2) Information Entropy Reduction.

### A.1  ATTENTION DYNAMICS AND CONTEXTUAL ANCHORING

In Transformer-based architectures, the probability of generating the next token $y_t$ is governed by the attention mechanism. Let the input sequence be the concatenation of the context $\mathcal{C}$ and the generated prefix $y_{<t}$. The attention output $\mathbf{h}_t$ at step $t$ is a weighted sum of value vectors:

$$\mathbf{h}_t = \underbrace{\sum_{j \in \mathcal{C}} \alpha_{t,j} \mathbf{v}_j}_{\text{Contextual Attention}} + \underbrace{\sum_{k \in y_{<t}} \alpha_{t,k} \mathbf{v}_k}_{\text{Parametric Attention}} \tag{2}$$

where $\alpha_{t,\cdot}$ represents the softmax-normalized attention weights. Hallucinations typically occur when the model fails to attend to the context (low $\sum_{j \in \mathcal{C}} \alpha_{t,j}$) and instead relies on internal parametric priors activated by the generated history.

**The Anchoring Effect:** We posit that Copy-Paste leverages the *Induction Head* mechanism (Olsson et al., 2022; Huang et al., 2025a), a circuit responsible for in-context copying. If we enforce the previous token $y_{t-1}$ to be a direct copy of a token $c_k \in \mathcal{C}$ (where $c_k$ is the token at position $k$ in the context), the query vector $\mathbf{q}_t$ (derived from $y_{t-1}$) will strongly correlate with the key vector $\mathbf{k}_{c_k}$ in the context.

Mathematically, maximizing the copying likelihood in Stage 1 (Section 4.1.1) effectively optimizes the attention weights such that:

$$\text{score}(\mathbf{q}_t, \mathbf{k}_{c_k}) \propto \mathbf{q}_t^\top \mathbf{k}_{c_k} \gg \mathbf{q}_t^\top \mathbf{k}_{\text{para}} \tag{3}$$

This creates a "Semantic Anchor," forcing the attention distribution $\alpha$ to collapse onto the context:

$$\lim_{\text{copying} \to \max} \sum_{j \in \mathcal{C}} \alpha_{t,j} \approx 1 \tag{4}$$

By ensuring $y_{t-1}$ is a copy, we mechanically guide the induction heads to retrieve the subsequent ground-truth token $c_{k+1}$ from $\mathcal{C}$, thereby physically suppressing the attention pathways that lead to parametric hallucinations. This aligns with our empirical observation in Figure 3, where Copy-PasteLLM exhibits significantly suppressed parametric logits.

### A.2  ENTROPY REDUCTION AND SEARCH SPACE CONSTRICTION

From an information-theoretic perspective, faithfulness hallucinations arise from high uncertainty in the conditional distribution $P(y_t|\mathcal{C}, y_{<t})$. Let $\mathcal{V}$ be the full vocabulary of the LLM, and $\mathcal{V}_\mathcal{C} \subset \mathcal{V}$ be the subset of tokens present in the context.

The Copy-Paste objective imposes a constraint that the generated response $Y$ must maximize lexical overlap with $\mathcal{C}$. This effectively serves as a regularization term that constrains the search space $\Omega$ from the vast $\mathcal{V}$ to the much smaller $\mathcal{V}_\mathcal{C}$.

The conditional entropy of the generation step without constraints is:

$$H(Y|\mathcal{C})_{\text{Base}} = -\sum_{w \in \mathcal{V}} P_{\text{Base}}(w|\mathcal{C}) \log P_{\text{Base}}(w|\mathcal{C}) \tag{5}$$

In standard generation, the probability mass is often distributed over a long tail of semantically similar but extrinsically hallucinated tokens from parametric memory. In contrast, CopyPasteLLM is optimized to concentrate probability mass on $\mathcal{V}_\mathcal{C}$:

$$\sum_{w \in \mathcal{V}_\mathcal{C}} P_{\text{CP}}(w|\mathcal{C}) \to 1 \implies P_{\text{CP}}(w \notin \mathcal{V}_\mathcal{C}) \to 0 \tag{6}$$

Consequently, the entropy of the Copy-Paste distribution is strictly lower than that of the base distribution:

$$H(Y|\mathcal{C})_{\text{CP}} \ll H(Y|\mathcal{C})_{\text{Base}} \tag{7}$$

By minimizing the entropy and pruning the probability of tokens $w \notin \mathcal{V}_\mathcal{C}$, we statistically minimize the risk of sampling hallucinated content. This theoretical result explains why our method, despite relying on lexical proxies, effectively improves semantic faithfulness by eliminating the "lexical pathways" that allow parametric priors to leak into the generation.

## B  EXPERIMENTAL SETUP

**Datasets**   We evaluate across four QA datasets: RAGTruth (Niu et al., 2024), a RAG hallucination corpus with 18K word-level annotated LLM responses; FaithEval (Ming et al., 2025), a counterfactual benchmark for contextual faithfulness; PubMedQA (Jin et al., 2019), a biomedical QA dataset where contexts contain 21% numeric descriptions; and ConFiQA (Bi et al., 2025), which includes both counterfactual and original contexts with gold answers. Table 4 summarizes the datasets and their roles (Train or Eval) across research questions (see Section 4).

Table 4: Datasets and their roles across 3 research questions. **Train** refers to the number of samples utilized for training our CopyPasteLLM, and **Eval** refers to the number of samples used for evaluation. The 20,000 samples of the PubMedQA Artificial subset were randomly sampled using the random seed 42 from the 211k entries.

| Dataset | Subset | Domain | Size | Gold Answer | RQ1 | RQ2 | RQ3 |
|---|---|---|---|---|---|---|---|
| RAGTruth | QA | Daily-Life | 839 | ✗ | Eval | only Train (16) | Eval |
| FaithEval | Counterfactual | Science | 1,000 | ✓ | Eval | Train / Eval (241 / 759) | Eval |
| PubMedQA | Labeled | Biomedicine | 1,000 | ✓ | Eval | Train / Eval (108 / 892) | Eval |
| PubMedQA | Artificial | Biomedicine | 20,000 | ✓ | - | Eval | - |
| ConFiQA | Counterfactual & Original | Wikidata | 36,000 | ✓ | - | Eval | - |

**Metrics**   For RQ1, we evaluate responses across multiple dimensions: contextual faithfulness using AlignScore (Zha et al., 2023) for overall answer assessment and MiniCheck (Tang et al., 2024) for sentence-level evaluation; hallucination detection via LLM-as-Judge (Qwen3-32B reasoning (Qwen-Team, 2025)) with pairwise comparisons (Zheng et al., 2023)) to identify Twist and Causal hallucinations (prompts detailed in Appendix L.3); response fluency measured by perplexity under GPT-2; copying behavior quantified through copy coverage ($\kappa$) and copy density ($\delta$); and query relevance assessed via Qwen3-Embedding-8B (Zhang et al., 2025b). For RQ2, we employ Hit Rate (following Li et al. (2025a)) and Accuracy, both requiring gold answers. Hit Rate measures the extent to which methods recognize contextual knowledge presence using Chain-of-Thought (CoT) prompting (Wei et al., 2022)), while Accuracy evaluates the degree of belief in contextual knowledge using direct answer prompting (prompts detailed in Appendix L.4). FaithEval provides ready-to-use multiple-choice options, whereas ConFiQA offers only Counterfactual and Original answers. For ConFiQA, we designate Counterfactual answers as correct in counterfactual contexts and Original answers as correct in original contexts. To increase task difficulty, we introduce an "unknown" option, allowing methods to express uncertainty when appropriate.

**Models & Baselines** We conduct experiments using four popular open-source LLMs as base models: *Mistral-7B-Instruct-v0.2 (M-7B)*, *Llama-3.1-8B-Instruct (L-8B)*, *Qwen2.5-72B-Instruct (Q-72B)*, and *DeepSeek-V3-0324 (D-V3)*. Copy-Paste-Prompting methods are evaluated on the four models. CopyPasteLLM is trained on M-7B, L-8B, and its predecessor LLaMA-3-8B-Instruct to enable comparison with more baselines.

**Stage 1 Baselines:** For Copy-Paste-Prompting evaluation, we compare against *Attributed* (Zhou et al., 2023) and *Citations*—the former a standard RAG approach, the latter requiring LLM-generated citations during abstractive generation (Zhang et al., 2023)). These methods serve dual purposes: validating our prompting effectiveness and **providing rejected responses for DPO training**.

**Stage 2 Baselines:** For CopyPasteLLM evaluation, we benchmark against state-of-the-art methods including prompting-based *Attributed*, Fine-tuning-based *Context-DPO* (Bi et al., 2025), Canoe (Si et al., 2025) and *ParamMute* (Huang et al., 2025b), and decoding-based *CoCoLex* (T.y.s.s et al., 2025)—a copy-based confidence decoding strategy for legal text faithfulness.

## C   COPY-PASTE-PROMPTING ANALYSIS

In this section, we provide a comprehensive analysis of the behavior and performance of our proposed Copy-Paste-Prompting methods (CP-Order, CP-Link, and CP-Refine). We focus on copying behavior and query Relevance.

### C.1   COPYING DEGREE ANALYSIS

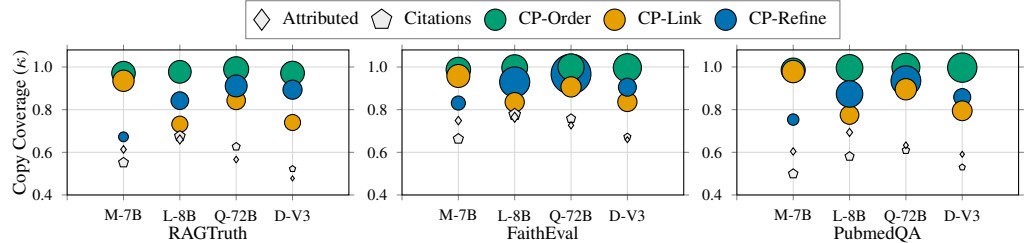

Figure 5: Copying degree across models and datasets. Copy-Paste-Prompting methods significantly outperform baselines in $\kappa$ and $\delta$ (area of point). Notably, the copying degree varies by dataset nature (FaithEval > PubMedQA > RAGTruth) and model capacity, with DeepSeek-V3 balancing copying and query relevance effectively.

Figure 5 illustrates the copying degree, measured by Copy Coverage ($\kappa$) and Copy Density ($\delta$), across different models and datasets. We observe three key trends:

**Superiority of Copy-Paste Methods:** All three Copy-Paste-Prompting variants consistently achieve significantly higher copying degrees compared to the *Attributed* and *Citations* baselines. This confirms that our prompting strategies successfully enforce the lexical reuse of context, which is the prerequisite for our subsequent preference learning pipeline.

**Model-Dependent Copying Behavior:** The ability to adhere to high-copying constraints varies by model size and intelligence. *Mistral-7B-Instruct-v0.2* generally exhibits the lowest copying degree among the models evaluated. This suggests that smaller models may struggle to maintain strict lexical constraints while simultaneously managing coherence. *DeepSeek-V3*, despite being the largest and most capable model, often shows the second-lowest copying degree among our methods (particularly in RAGTruth and PubMedQA). We hypothesize that this is due to the model's advanced capability to balance conflicting objectives; rather than blindly maximizing copying at the expense of fluency or logic, DeepSeek-V3 likely optimizes for a "sweet spot" that maintains high copying while ensuring the response remains natural and logically sound.

**Dataset-Specific Characteristics:** We observe a distinct ordering in copying magnitude across datasets: `FaithEval > PubMedQA > RAGTruth`. (1)`FaithEval (Highest Copying):`

Since this dataset focuses on counterfactual robustness, the parametric knowledge is intentionally incorrect. Models are forced to rely entirely on the provided context to answer correctly, leading to maximum copying. (2) `PubMedQA:` The biomedical domain involves specific terminology and factual definitions that are difficult to paraphrase without losing precision, naturally encouraging higher lexical reuse. (3) `RAGTruth (Lowest Copying):` This dataset involves open-ended, real-world queries that often necessitate abstractive synthesis and summarization. Although its overall copying degree is lower than the other two domain-specific datasets, our Copy-Paste methods still effectively enforce substantial and useful lexical reuse from the context.

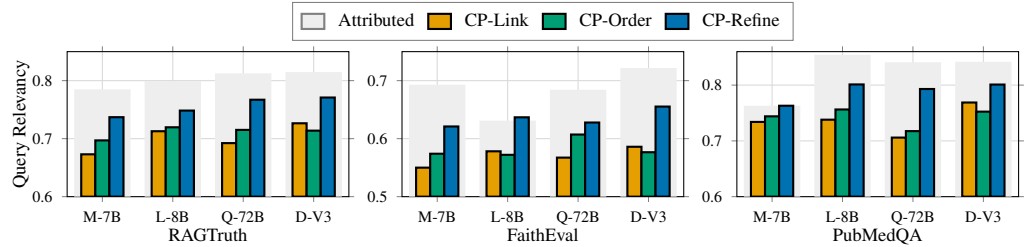

Figure 6: Query relevancy performance. CP-Refine consistently yields the most relevant responses. The efficacy of CP-Link is model-dependent; only the highly capable DeepSeek-V3 utilizes the linking mechanism to improve relevance over the rigid CP-Order approach.

## C.2 Query Relevancy Analysis

As shown in Figure 6, CP-Refine consistently achieves the highest query relevancy among the three proposed methods. This validates the effectiveness of the "Reviewer" component in our soft-constraint loop, which explicitly critiques and guides the "Writer" to address the query while maintaining copying. The performance of CP-Link is strongly correlated with model intelligence. While CP-Link is designed to improve upon CP-Order by adding transitional text, only the most capable model, `DeepSeek-V3`, successfully leverages this freedom to enhance query relevance over CP-Order. Smaller models (e.g., Mistral-7B) often fail to generate meaningful transitions, resulting in performance similar to or lower than the strict CP-Order method.

## D Implementation Details

### D.1 CopypPaste-Prompting

---

**Algorithm 1** Copy-Paste-Prompting: Constructing High-Copying Responses

---

**Require:** Query $Q$, Context $C$, Method $M \in \{$CP-Order, CP-Link, CP-Refine$\}$, Threshold $\theta_\sigma$, Max iterations $T_{\max}$
**Ensure:** High-copying response $A$
1: **if** $M \in \{$CP-Order, CP-Link$\}$ **then**                                    ▷ Hard-constraint methods
2:     $\{s_1, ..., s_n\} \leftarrow$ ExtractRelevantSentences$(C, Q)$               ▷ Common extraction step
3:     **if** $M =$ CP-Order **then**                                               ▷ Direct sentence ordering
4:         $A \leftarrow$ DirectOrdering$(\{s_i\}, Q)$
5:     **else**                                                        ▷ CP-Link: Ordering via transition generation
6:         $A \leftarrow$ GenerateTransitionsWithOrdering$(\{s_i\}, Q)$
7:     **end if**
8: **else**                                                       ▷ CP-Refine: Soft-constraint with iterative refinement
9:     $A^{(0)} \leftarrow$ Writer$(Q, C), t \leftarrow 0$
10:     **while** $t < T_{\max}$ **or** $\sigma^{(t)} < \theta_\sigma$ **do**          ▷ Until copying score meets threshold
11:         feedback $\leftarrow$ Reviewer$(A^{(t)}, Q, C)$          ▷ Verbal supervision on relevance & fluency
12:         $\sigma^{(t)} \leftarrow \alpha \cdot \kappa(A^{(t)}, C) + \min(\delta(A^{(t)}, C)^\beta/\gamma, \varepsilon)$                          ▷ Copy score
13:         **if** $\sigma^{(t)} \geq \theta_\sigma$ **then**                         ▷ Hard constraint on copying score only
14:             **break**
15:         **end if**
16:         $A^{(t+1)} \leftarrow$ Writer$(Q, C, $feedback$), t \leftarrow t + 1$
17:     **end while**
18:     $A \leftarrow A^{(t)}$
19: **end if**
20: **return** $A$

---

The Copy-Paste-Prompting methods were implemented using a modular pipeline architecture. For the `CP-Order` method, we employed a text similarity threshold of 0.95 for validating extracted sentences against the source context, utilizing both direct string matching and fuzzy matching with sliding windows to ensure strict extraction accuracy. The `CP-Link` method generates transition sentences with a constraint of no more than 15 words to maintain conciseness while ensuring logical flow. For the `CP-Refine`, we utilized an iterative refinement process with a dual-agent system (Writer and Reviewer) implemented via LangGraph[1]. This process was configured with a maximum of $T_{\max} = 5$ iterations and a target copying score threshold $\theta_\sigma = 0.99$. The composite copying score $\sigma^{(t)}$ is calculated using hyperparameters $\alpha = 0.6$, $\beta = 0.25$, $\gamma = 4$, and $\varepsilon = 0.4$, effectively balancing the contribution of copy coverage ($\kappa$) and copy density ($\delta$) to guide the generation toward high contextual fidelity. All prompting methods were run with a low temperature setting (0.1) to ensure consistent outputs.

## D.2 COPYPASTELLM

---

**Algorithm 2** CopyPasteLLM: Automated Preference Construction and Training

---
**Require:**
1: Query-context pairs $\{(Q_i, C_i)\}_{i=1}^N$;
2: Methods $\mathcal{T} = \{$Base, Attributed, Citations, CP-Order, CP-Link, CP-Refine$\}$;
3: Metrics $\{f_j, \theta_j\}_{j=1}^6$; Temperature $\beta$
**Ensure:** Trained model $\pi_\theta$ with internalized contextual belief
4: Initialize $\mathcal{D} \leftarrow \emptyset$
5: **for** each $(Q_i, C_i)$ **do**
6:   $\mathcal{R}_i \leftarrow \{$GenerateResponse$(Q_i, C_i, m) : m \in \mathcal{T}\}$         ▷ Generate candidates
7:   $\mathcal{R}_i^f \leftarrow \{r \in \mathcal{R}_i : \bigwedge_{j=1}^6 (f_j(r) \bowtie_j \theta_j)\}$         ▷ Multi-criteria filtering
8:   ratings $\leftarrow$ EloTournament$(\mathcal{R}_i^f, C_i)$       ▷ Pairwise LLM-as-Judge with Elo scoring
9:   $r_i^* \leftarrow \arg\max_{r \in \mathcal{R}_i^f}$ ratings$[r]$          ▷ Select best response
10:   **if** $A_i^{\text{gold}}$ and $A_i^{\text{wrong}}$ available **then**      ▷ Handle samples with answer annotations
11:    $r_i^{\text{chosen}} \leftarrow r_i^* \oplus A_i^{\text{gold}}$    ▷ Append gold answer to transform reasoning into conclusion
12:    $\mathcal{R}_i^{\text{rejected}} \leftarrow \{r \oplus A_i^{\text{wrong}} : r \in \mathcal{R}_i^f \cap \{$CP-Order, CP-Link, CP-Refine$\} \setminus \{r_i^*\}\}$   ▷ Append wrong answers to other CP methods
13:    $\mathcal{D} \leftarrow \mathcal{D} \cup \{(Q_i \oplus C_i, r_i^{\text{chosen}}, r^-) : r^- \in \mathcal{R}_i^{\text{rejected}} \cup (\mathcal{R}_i^f \setminus \{$CP methods$\})\}$
14:   **else**              ▷ Handle samples without answer annotations
15:    $\mathcal{D} \leftarrow \mathcal{D} \cup \{(Q_i \oplus C_i, r_i^*, r^-) : r^- \in \mathcal{R}_i^f \setminus \{r_i^*\}\}$   ▷ Use original responses without answer appending
16:   **end if**
17: **end for**
18: Initialize $\theta, \pi_{\text{ref}}$         ▷ DPO training with $5N$ preference pairs from $N$ samples
19: **while** not converged **do**
20:   **for** each $(x, y_w, y_l) \in \mathcal{D}$ **do**    ▷ Leverage 5× data efficiency: each sample yields 5 preference pairs
21:    $\mathcal{L} = -\log \sigma \left( \beta \log \frac{\pi_\theta(y_w|x)}{\pi_{\text{ref}}(y_w|x)} - \beta \log \frac{\pi_\theta(y_l|x)}{\pi_{\text{ref}}(y_l|x)} \right)$
22:    Update $\theta$ using $\nabla_\theta \mathcal{L}$
23:   **end for**
24: **end while**
25: **return** $\pi_\theta$

---

We fine-tune CopyPasteLLM on three instruction-tuned bases—`Mistral-7B-Instruct-v0.2`[2], `LLaMA-3-8B-Instruct`[3], and `Llama-3.1-8B-Instruct|using`[4]. Direct Preference Optimization (DPO) with parameter-efficient LoRA adapters, based on responses generated by DeepSeek-V3-0324. We adapt attention and MLP projections (`q_proj`, `k_proj`, `v_proj`, `o_proj`, `gate_proj`, `up_proj`, `down_proj`) with $r = 64$, $\alpha = 128$, and `dropout=0`. Training uses a maximum prompt length of 8192 and a maximum generation length of 1024; the per-device batch size is 2, combined with 8 gradient-accumulation steps. We optimize with AdamW (learning rate 5e-5, weight decay 0.01, max gradient norm 1.0) under a cosine schedule with a 5% warmup and no label smoothing; the DPO temperature is set to $\beta = 0.3$. To balance compute and convergence, we train for 2 epochs on Mistral-7B-Instruct-v0.2 and LLaMA-3-8B-Instruct, and for 1 epoch on Llama-3.1-8B-Instruct.

---

[1]https://github.com/langchain-ai/langgraph

[2]https://huggingface.co/mistralai/Mistral-7B-Instruct-v0.2

[3]https://huggingface.co/meta-llama/Meta-Llama-3-8B-Instruct

[4]https://huggingface.co/meta-llama/Llama-3.1-8B-Instruct

# E    TASK-SPECIFIC PERFORMANCE EVALUATION OF COPYPASTELLM

To gain a deeper understanding of CopyPasteLLM's robustness and versatility, we conduct a fine-grained performance analysis across three dimensions: conflict complexity (ConFiQA), knowledge domains (FaithEval), and reasoning ambiguity (PubMedQA).

## E.1    IMPACT OF CONFLICT COMPLEXITY AND MULTI-HOP REASONING

We utilize the three subsets of the ConFiQA (Bi et al., 2025) dataset to evaluate how CopyPasteLLM handles increasing levels of reasoning complexity and counterfactual conflict density:

- ConFiQA-QA (Question-Answering): Represents single-hop reasoning with a single point of knowledge conflict.
- ConFiQA-MR (Multi-hop Reasoning): Involves multi-hop structures where only one step contains a knowledge conflict, testing the model's ability to integrate counterfactuals into a reasoning chain.
- ConFiQA-MC (Multi-Conflicts): The most challenging setting, featuring multi-hop structures where *all* reasoning steps are modified to be counterfactual, creating a global knowledge conflict.

As shown in Table 1, CopyPasteLLM demonstrates exceptional robustness as task difficulty increases. On the ConFiQA-MC subset, where models must adhere to a completely counterfactual reality, baseline performance typically collapses. For instance, on Llama-3.1-8B, the *Attributed* method achieves only 15.5% accuracy. CopyPasteLLM maintains a remarkable accuracy of **83.5%**, strictly follows the instruction of generating responses based on context, and significantly mitigates internal parametric resistance. Notably, on the Mistral-7B backbone, CopyPasteLLM (trained on only 365 samples) achieves **82.5%** on the MC subset, outperforming even the *Context-DPO* method (80.4%) which was trained on 18,000 samples seen during training.

## E.2    PERFORMANCE ACROSS DIVERSE KNOWLEDGE TYPES

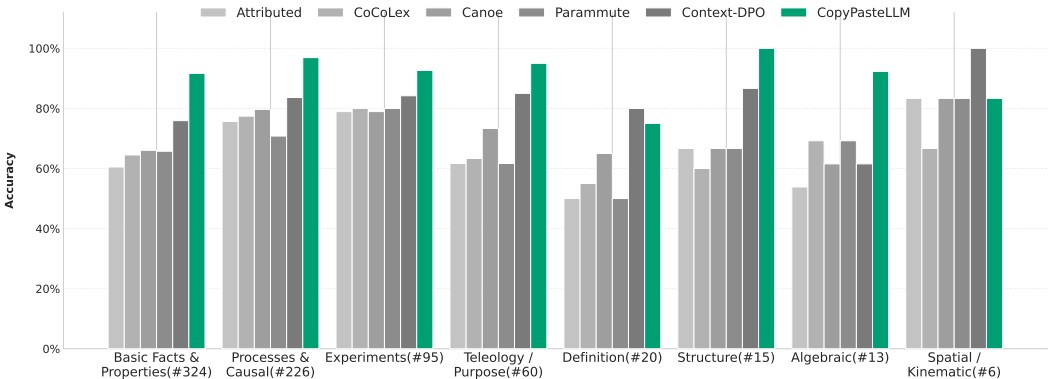

Figure 7: Performance comparison across diverse knowledge domains. CopyPasteLLM consistently outperforms or remains highly competitive against strong baselines across most categories, demonstrating robustness in both factual (e.g., *Basic Facts*) and reasoning-intensive domains (e.g., *Processes & Causal*, *Experiments*).

To analyze performance across different knowledge domains, we classify the FaithEval-Counterfactual (Ming et al., 2025) dataset samples into eight distinct knowledge types (e.g., Basic Facts, Processes, Experiments). Since the original dataset lacks these labels, we employed a strong model, DeepSeek-V3.1, to classify the samples based on the taxonomy defined in the ARC-Challenge (Clark et al., 2018). We utilized a Chain-of-Thought (CoT) prompting strategy with majority voting (temperature sampling over 5 runs) to ensure classification reliability. Figure 7 illustrates the accuracy breakdown. CopyPasteLLM (green bar) consistently outperforms or matches the strongest baselines across all categories.

### E.3 ROBUSTNESS TO REASONING AMBIGUITY

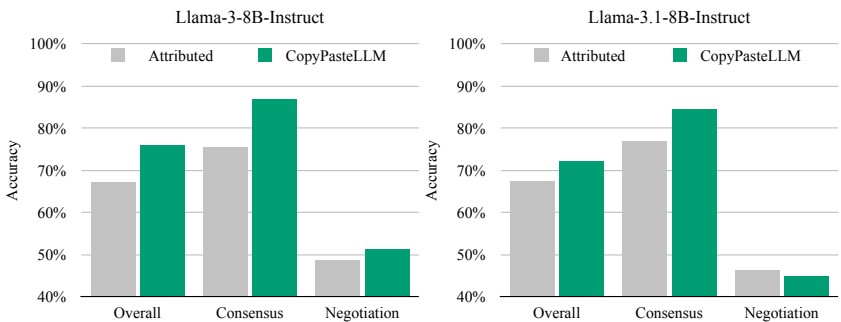

Figure 8: Performance breakdown on PubMedQA-Labeled by reasoning difficulty. Accuracy is compared between Attributed and CopyPasteLLM models across the Consensus (clear evidence) and Negotiation (ambiguous context) subsets, demonstrating the highest gains in samples with explicit evidence.

We further analyze performance on the PubMedQA-Labeled (Jin et al., 2019) dataset, distinguishing between samples based on reasoning setting as defined by the original dataset annotators:

- Consensus: Samples where independent experts agreed on the answer easily, implying the context provided clear, unambiguous evidence.

- Negotiation: Samples where experts initially disagreed and had to negotiate a final answer, implying the context was ambiguous, implicit, or difficult to interpret.

Figure 8 presents the accuracy on these splits. We observe a distinct pattern:

- **Clear Evidence (Consensus):** CopyPasteLLM delivers significant improvements. For Llama-3-8B, accuracy improves from 75.49% (Attributed) to 86.85% (CopyPasteLLM); for Llama-3.1-8B, it improves from 76.79% to 84.58%. This confirms that when explicit evidence exists, our method effectively anchors the model to it.

- **Ambiguous Evidence (Negotiation):** The performance gains are more modest or neutral. For Llama-3-8B, the performance has slightly improved (48.91% vs 51.27%), while for Llama-3.1-8B, its performance is slightly lower than the baseline (46.38% vs 44.93% ). This result is expected and rational: the Copy-Paste pattren relies on the existence of text fragments that directly support the answer. In "Negotiation" samples, where the evidence is implicit or ambiguous, there may be no clear fragments to copy that definitively resolves the query.

Figures 9 and 10 compare the accuracy of the `Attributed` and `CopyPasteLLM` across the top frequent MeSH terms, demonstrating consistent performance gains in handling specific medical topics by enforcing contextual adherence.

## F RESPONSE LENGTH ANALYSIS

To further investigate the behavioral characteristics of CopyPasteLLM under the Chain-of-Thought (CoT) setting, we conduct a granular analysis of response length, copying degree, and the semantic relevance of copied content. The statistical comparison against five baselines (based on LLaMA-3-8B-Instruct) is presented in Table 5.

**Response Length and Reasoning Capability.** As shown in Table 5, there is a significant divergence in response lengths among methods. *ParamMute* (Huang et al., 2025b) exhibits an extremely short median length of 4.0 tokens. This indicates a failure to adhere to the CoT instructions; instead of generating a reasoning chain, it tends to output the final answer option directly, potentially explaining its suboptimal performance in complex reasoning tasks. Conversely, abstractive baselines

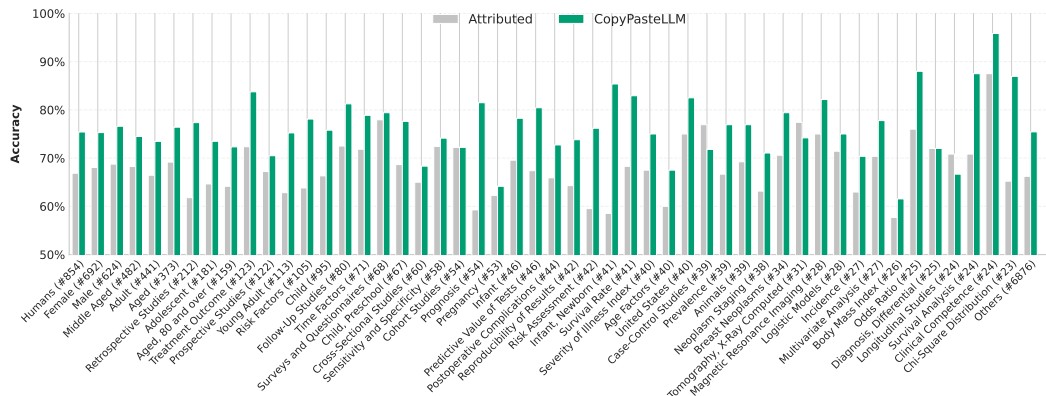

Figure 9: Domain-specific performance analysis of CopyPasteLLM (based on Llama-3-8b-instruct) on PubMedQA, categorized by Medical Subject Headings (MeSH).

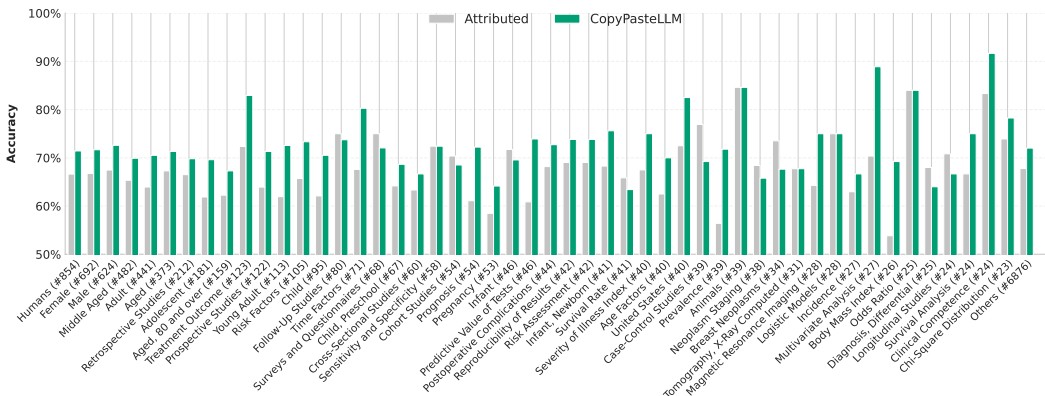

Figure 10: Domain-specific performance analysis of CopyPasteLLM (based on Llama-3.1-8b-instruct) on PubMedQA, categorized by Medical Subject Headings (MeSH).

Table 5: Response Analysis by Method under CoT setting (Base: LLaMA-3-8B-Instruct). Metrics reported are Median ± Standard Deviation.

| Method | Length | $\kappa$ (Coverage) | $\delta$ (Density) |
|---|---|---|---|
| Attributed (Zhou et al., 2023) | 199.0 ± 68.29 | 0.552 ± 0.129 | 2.70 ± 2.85 |
| CoCoLex (T.y.s.s et al., 2025) | 75.0 ± 40.01 | 0.989 ± 0.040 | 50.08 ± 37.68 |
| Canoe (Si et al., 2025) | 198.0 ± 69.30 | 0.551 ± 0.126 | 2.67 ± 3.68 |
| Context-DPO (Bi et al., 2025) | 159.5 ± 62.72 | 0.631 ± 0.130 | 4.17 ± 4.32 |
| ParamMute (Huang et al., 2025b) | 4.0 ± 18.69 | 1.000 ± 0.182 | 3.00 ± 13.23 |
| **CopyPasteLLM (Ours)** | 126.0 ± 78.71 | 0.844 ± 0.135 | 10.49 ± 15.49 |

like *Attributed* and *Canoe* generate longer responses ($\approx 198$ tokens) but with lower copy density ($\delta \approx 2.7$), suggesting a reliance on internal parametric knowledge for generation. *CopyPasteLLM* maintains a moderate and sufficient response length (126.0 tokens), striking a balance that allows for adequate reasoning steps while enforcing grounding through copying.

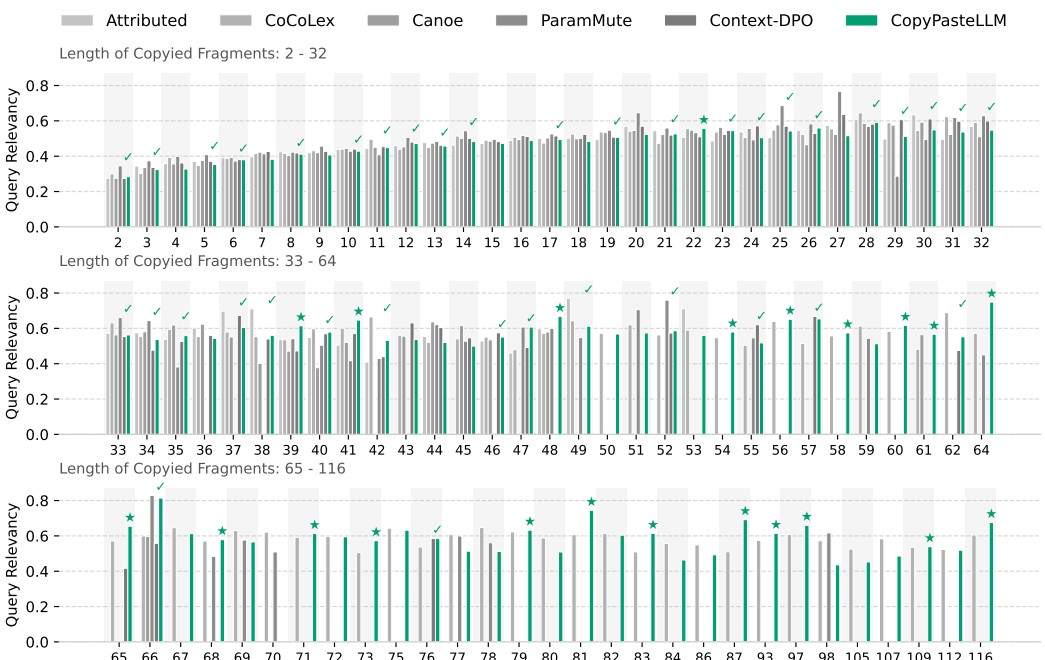

Figure 11: Query Relevancy analysis of copied fragments across different length buckets. The height of the bars represents the average cosine similarity between the copied fragments and the query. CopyPasteLLM (green) maintains competitive relevance across all lengths.

**Rational vs. Blind Copying.** Analyzing the copying metrics reveals distinct strategies. *Co-CoLex* (T.y.s.s et al., 2025) achieves a near-perfect copy coverage ($\kappa \approx 0.989$) and an exceptionally high copy density ($\delta \approx 50.08$). However, combined with its low Hit Rate of 17.9% on FaithEval (refer to Table 1), this suggests a tendency towards "blind copying"—reproducing the entire context verbatim without selective filtering or logical reasoning. In contrast, *CopyPasteLLM* demonstrates a high but rational copying behavior ($\kappa \approx 0.844$, $\delta \approx 10.49$). It copies significantly more continuous spans than standard baselines (Context-DPO's $\delta \approx 4.17$) to ensure faithfulness, yet avoids the unselective copying observed in CoCoLex.

**Query Relevancy of Copied Fragments.** To verify that CopyPasteLLM copies *meaningful* evidence rather than irrelevant noise, we analyze the semantic relevance of copied fragments in Figure 11. We extracted all copied fragments with a length $\geq 2$ using Algorithm 3 and calculated their cosine similarity with the input query using text embedding model. The results indicate that across various fragment lengths (ranging from short phrases to long sentences), CopyPasteLLM maintains a consistent and high level of query relevancy (comparable to or exceeding baselines). This confirms that our method effectively identifies and copies context segments that are semantically pertinent to the user's question.

## G   ABLATION STUDY AND TRAINING DYNAMICS ANALYSIS

To rigorously dissect the contribution of each component in CopyPasteLLM and evaluate its training stability, we conducted ablation studies based on the Llama-3.1-8B-Instruct. We utilized the counterfactual subsets of ConFiQA (QA, MR, and MC) as the testset. All models were trained on the same 365 preference pairs for 2 epochs (218 steps). To ensure statistical significance, we evaluated

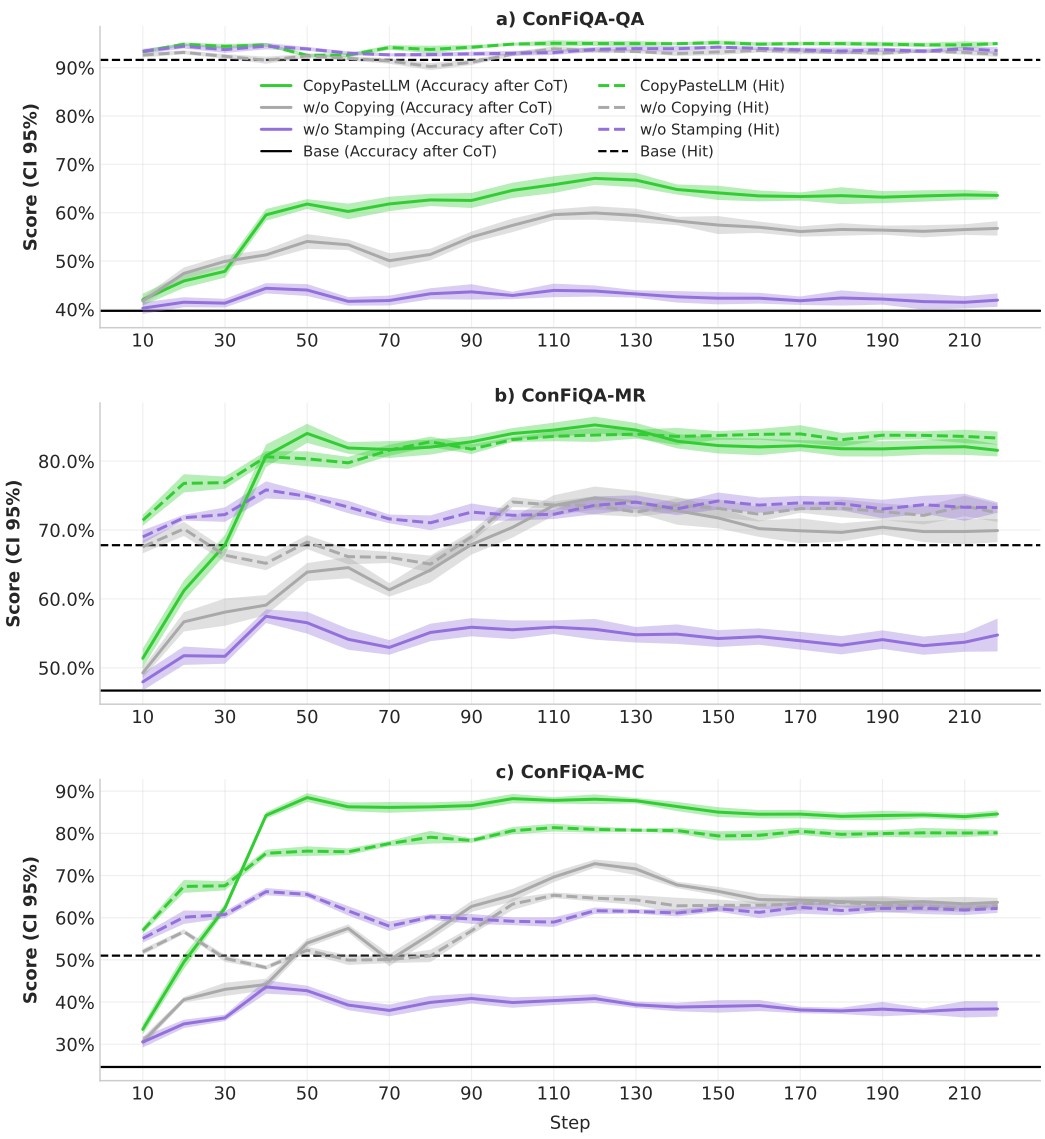

Figure 12: Ablation study and training dynamics on ConFiQA datasets. Solid lines represent *Accuracy after CoT* (strict JSON format, see Prompt L.6), and dashed lines represent *Hit Rate* (see Prompt L.4). Shaded areas indicate 95% confidence intervals across 8 random seeds. Copy-PasteLLM (Green) consistently outperforms variants without high-copying data (Grey) and without answer stamping (Purple).

the models every 10 steps using 8 different random seeds ($0 - 6, 42$) with a temperature of 0.7 and top-p of 0.95. The results, visualized with 95% confidence intervals (calculated via t-distribution), are presented in Figure 12.

## G.1 IMPACT OF HIGH-COPYING PREFERENCE DATA (W/O COPYING)

To assess the necessity of our specific high-copying response construction, we implemented the **w/o Copying** variant (grey lines). In this setting, we excluded the three Copy-Paste-Prompting methods (CP-Order, CP-Link, CP-Refine) and constructed preference pairs solely from standard baselines (Base, Attributed, Citations), selecting the top 365 samples based on multi-criteria filtering (see Figure 2).

As shown in Figure 12, the full **CopyPasteLLM** (green lines) consistently outperforms the w/o Copying variant across all three datasets, particularly in the more challenging *Accuracy after CoT* metric (solid lines). Even when standard RAG responses are filtered for high faithfulness, they lack the explicit lexical anchoring provided by our Copy-Paste. The performance gap indicates that the specific structural characteristic of CopyPaste—not just semantic correctness—is a stronger supervision signal for suppressing parametric hallucinations (refer to Appendix A for mechanistic interpretation of Copy-Paste). The high-copying preference data effectively teaches the model to prioritize the retrieved context over internal knowledge.

## G.2 SIGNIFICANCE OF ANSWER STAMPING (W/O STAMPING)

We evaluated the role of the *Stamping Answers* step (Stage 2, Step 3 in Figure 2) by removing it, denoted as **w/o Stamping** (purple lines). In this variant, the model learns from the chosen response's reasoning trace without the explicit appending of the ground-truth answer label.

The results reveal a critical insight: removing stamping leads to a drastic performance drop, often falling close to Base model (black horizontal line), especially on the *Accuracy* metric. The *Accuracy after CoT* metric requires the model to output a strict JSON format {`"reasoning":` `"..."`, `"answer":` `"..."`}. Without stamping, the DPO optimization primarily aligns the reasoning style but fails to enforce a definitive commitment to the correct conclusion. The huge gap between the solid purple line (Accuracy) and dashed purple line (Hit) in ConFiQA-QA (Figure 12a) suggests that while the model might mention the correct entity (Hit), it struggles to formalize it as the final answer without the explicit "conclusion-forcing" signal provided by stamping.

## G.3 TRAINING DATA EFFICIENCY AND STABILITY

Figure 12 also illustrates the training dynamics over 218 steps.

**Rapid Convergence and Data Efficiency:** CopyPasteLLM exhibits remarkable learning efficiency. The performance curves generally rise sharply and reach a plateau around step 120–130 (approximately 1 epoch). Notably, on the ConFiQA-MR and ConFiQA-MC subsets (Figures 12 b/c), the model approaches its peak performance as early as step 50. This rapid saturation suggests that the high-copying preference signal provided by our method is highly potent, enabling the model to realign its internal belief mechanism with minimal data updates. The marginal utility of increasing data volume appears to diminish quickly, indicating that performance gains are driven primarily by the *quality* and *precision* of the constructed preference pairs rather than the sheer scale of the dataset.

**Robustness:** The narrow shaded areas (95% confidence intervals) across 8 random seeds indicate that our method is highly stable and reproducible. Unlike the w/o Stamping variant, which shows higher variance and instability (wider purple bands in Figure 12b/c), CopyPasteLLM consistently converges to a high-performance state regardless of generation randomness.

## H PERFORMANCE OF MAINSTREAM MODELS ON FAITHEVAL

The FaithEval counterfactual subset presents a challenging benchmark where mainstream LLMs demonstrate surprisingly low performance, with more powerful models often achieving lower ac-

curacy rates (see Table 6). This counterintuitive pattern suggests that larger models may rely more heavily on their parametric knowledge, leading to reduced contextual faithfulness when faced with counterfactual information.

Table 6: Performance comparison on FaithEval counterfactual subset. The table reports accuracy scores of mainstream models from the FaithEval (Ming et al., 2025)) alongside our CopyPasteLLM method evaluated on three 7-8B parameter models. **Bold** values indicate our best performing method, Underlined values indicate the second-best performing method and *Italic* values indicate the third-best performing method.

| Model | Accuracy (%) |
|---|---|
| Mistral-7B-Instruct-v0.3 | 73.8 |
| Llama-3.1-8B-Instruct | 68.5 |
| Llama-3-8B-Instruct | 66.5 |
| Mistral-Nemo-Instruct-2407 | 58.3 |
| gpt-3.5-turbo | 57.1 |
| Command R | 69.3 |
| Phi-3.5-mini-instruct | 66.8 |
| Command R+ | 73.6 |
| gemma-2-9b-it | 55.7 |
| gemma-2-27b-it | 55.7 |
| gpt-4o-mini | 50.9 |
| Phi-3-mini-128k-instruct | 75.7 |
| Phi-3-medium-128k-instruct | 60.8 |
| Llama-3.1-70B-Instruct | 55.2 |
| Llama-3-70B-Instruct | 60.5 |
| Claude 3.5 Sonnet | 73.9 |
| gpt-4-turbo | 41.2 |
| gpt-4o | 47.5 |
| CopyPasteLLM (Based on Llama-3-8B-Instruct) | **92.8** |
| CopyPasteLLM (Based on Mistral-7B-Instruct-v0.2) | *89.3* |
| CopyPasteLLM (Based on Llama-3.1-8B-Instruct) | 92.6 |

## I  COPY FRAGMENT DETECTION

The following copy fragment detection algorithm 3 is adapted from Grusky et al. (2018) and included here for completeness of this paper.

---

**Algorithm 3** Copy Fragment Detection

---

**Require:** Context sequence $C = [c_0, c_1, \ldots, c_{m-1}]$; Answer sequence $A = [a_0, a_1, \ldots, a_{n-1}]$.
**Ensure:** Set of copy fragments $\mathcal{F} = \{f_1, f_2, \ldots, f_k\}$
1: $\mathcal{F} \leftarrow \emptyset, i \leftarrow 0$
2: **while** $i < n$ **do**
3:     $\ell_{\max} \leftarrow 0, M \leftarrow \{j \mid j \in [0, m-1], c_j = a_i\}$           ▷ Find all matching positions in context
4:     **for** $m \in M$ **do**
5:         $\ell \leftarrow 0$
6:         **while** $i + \ell < n$ **and** $m + \ell < m$ **and** $a_{i+\ell} = c_{m+\ell}$ **do**
7:             $\ell \leftarrow \ell + 1$
8:         **end while**
9:         **if** $\ell > \ell_{\max}$ **then**
10:             $\ell_{\max} \leftarrow \ell$
11:         **end if**
12:     **end for**
13:     **if** $\ell_{\max} > 0$ **then**
14:         $\mathcal{F} \leftarrow \mathcal{F} \cup \{[a_i, a_{i+1}, \ldots, a_{\ell_{\max}-1}]\}$     ▷ Copy the matching subsequnces to fragment set
15:         $i \leftarrow i + \ell_{\max}$
16:     **else**
17:         $i \leftarrow i + 1$
18:     **end if**
19: **end while**
20: **return** $\mathcal{F}$

---

## J  ANALYSIS OF CONTEXT-PARAMETER COPYING CAPTURING

---

**Algorithm 4** Context-Parameter Copying Capturing

---

**Require:** Given string of context $C$ and query, the LLM generates a token answer $A_{\text{ctx}}$ of length $n$, $\mathcal{P}_i$: logits distribution of the $i$-th token, $H_i$: hidden states of the $i$-th token, $\mathcal{V}$: vocabulary of LLM. $A_{\text{para}}$: token answer generated without context, $K$: scope of knowledge capture.

**Ensure:** Captured knowledge logits and hidden states $P_{\text{ctx}}, P_{\text{para}}, H_{\text{ctx}}, H_{\text{para}}$

1: Initialize $P_{\text{ctx}}, P_{\text{para}}, H_{\text{ctx}}, H_{\text{para}} \leftarrow \emptyset, T_{\text{ctx}}, T_{\text{para}} \leftarrow \emptyset$         ▷ Token lists for captured tokens
2: $S_{\text{com}} = \text{commonSubstringMatching}(C, A_{\text{para}})$         ▷ Identify common substrings
3: **for** $i$ in $[1, 2, \ldots, n]$ **do**
4:     $\mathcal{P}_i' = \text{softmax}(\mathcal{P}_i)$         ▷ Normalize logits to probability distribution
5:     $\mathcal{V}_i' = \text{sort}(\mathcal{V}, \mathcal{P}_i')$         ▷ Sort vocabulary tokens by $\mathcal{P}_i'$ in descending order
6:     **for** $j$ in $[1, 2, \ldots, K]$ **do**         ▷ Only consider the top-K most likely tokens
7:         $x_j = \mathcal{V}_i'[j]$         ▷ Get $j$-th most probable token
8:         **if** isMeaningless($x_j$) **then continue**         ▷ Skip meaningless tokens, e.g. function words
9:         **end if**
10:         **if** $x_j$ in $S_{\text{com}}$ **then break**         ▷ $x_j$ is common to both context and parametric generation
11:         **end if**
12:         **if** $x_j$ in $C$ and $x_j \notin T_{\text{ctx}}$ **then**         ▷ Capture contextual knowledge token
13:             $P_{\text{ctx}} \leftarrow P_{\text{ctx}} \cup \{\mathcal{P}_{i,j}'\}, H_{\text{ctx}} \leftarrow H_{\text{ctx}} \cup \{H_i\}, T_{\text{ctx}} \leftarrow T_{\text{ctx}} \cup \{x_j\}$    **break**
14:         **end if**
15:         **if** $x_j$ in $A_{\text{para}}$ and $x_j \notin T_{\text{para}}$ **then**         ▷ Capture parametric knowledge token
16:             $P_{\text{para}} \leftarrow P_{\text{para}} \cup \{\mathcal{P}_{i,j}'\}, H_{\text{para}} \leftarrow H_{\text{para}} \cup \{H_i\}, T_{\text{para}} \leftarrow T_{\text{para}} \cup \{x_j\}$    **break**
17:         **end if**
18:     **end for**
19: **end for**
20: **return** $P_{\text{ctx}}, P_{\text{para}}, H_{\text{ctx}}, H_{\text{para}}$

---

This section provides comprehensive analysis of our Context-Parameter Copying Capturing algorithm across multiple datasets and model architectures. Figure 13 presents the complete logits power distribution analysis across all three datasets (RAGTruth, FaithEval, PubMedQA), revealing how CopyPasteLLM and base models differ in their reliance on contextual versus parametric knowledge throughout the generation process. Figures 14 and 15 complement the main text analysis by showing hidden states distributions on FaithEval and RAGTruth datasets, demonstrating the semantic separation between contextual and parametric knowledge representations in CopyPasteLLM compared to base models.

**Logits Power Calculation Formula**  We employ the following formula to calculate the logits power for each response token, measuring the model's reliance on contextual versus parametric knowledge during generation:

$$\text{logits\_power} = \left( \sum_{i=1}^{n} \ell_i^2 \right) \times \sqrt{n} \tag{8}$$

where $\ell_i$ denotes the logit value of the $i$-th token, and $n$ represents the number of samples in the dataset that have contextual or parametric knowledge at this position.

## K  LIMITATIONS AND FUTURE WORKS

While CopyPasteLLM demonstrates remarkable effectiveness in enhancing contextual faithfulness through high-copying behavior and achieves substantial performance improvements with exceptional data efficiency, several promising directions warrant future investigation.

**Incomplete Context Scenarios:** Our current framework assumes that the provided context contains sufficient information to answer the query. When context is incomplete or lacks relevant details, the copy-paste paradigm may struggle to generate satisfactory responses. Future work could explore adaptive mechanisms that dynamically assess context sufficiency and gracefully handle information gaps, potentially by incorporating uncertainty quantification or developing hybrid strategies that selectively combine contextual and parametric knowledge based on context completeness.

**Deeper Mechanistic Understanding:** While our Context-Parameter Copying Capturing algorithm provides valuable insights into logits and hidden state distributions, a more comprehensive mechanistic analysis could examine the roles of specific model components such as attention heads and

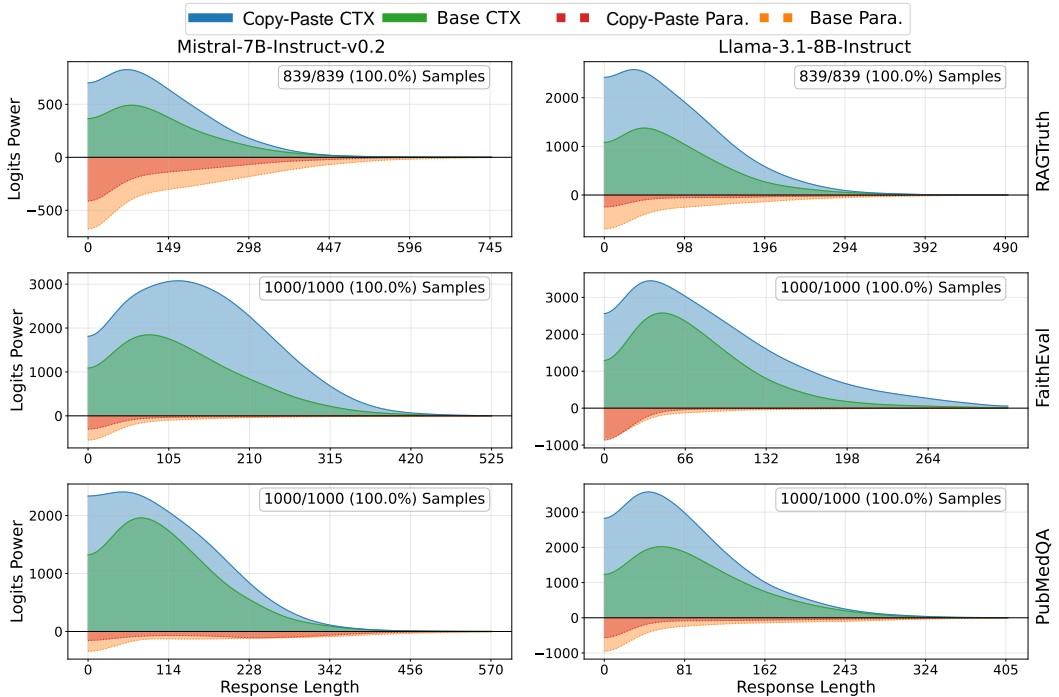

Figure 13: Logits power distribution across response lengths for contextual (CTX) and parametric (Para.) knowledge. Values above x=0 indicate CTX logits power, values below x=0 indicate Para. logits power (negated for visualization).

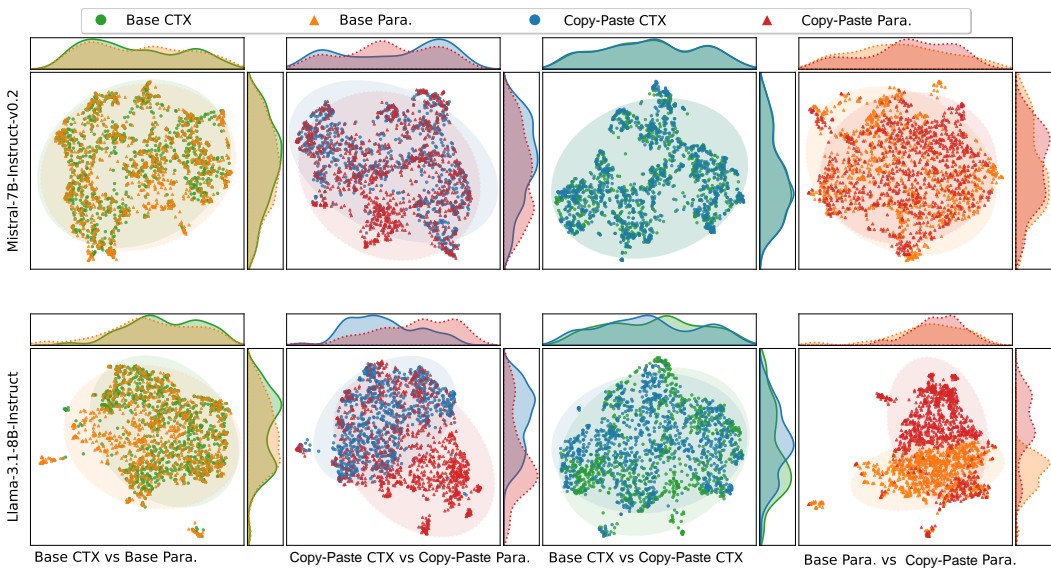

Figure 14: Dimensionality reduction visualization of hidden states distributions between contextual (CTX) and parametric (Para.) knowledge on FaithEval dataset across two base models. Each subplot shows pairwise comparisons with marginal KDE distributions and confidence ellipses.

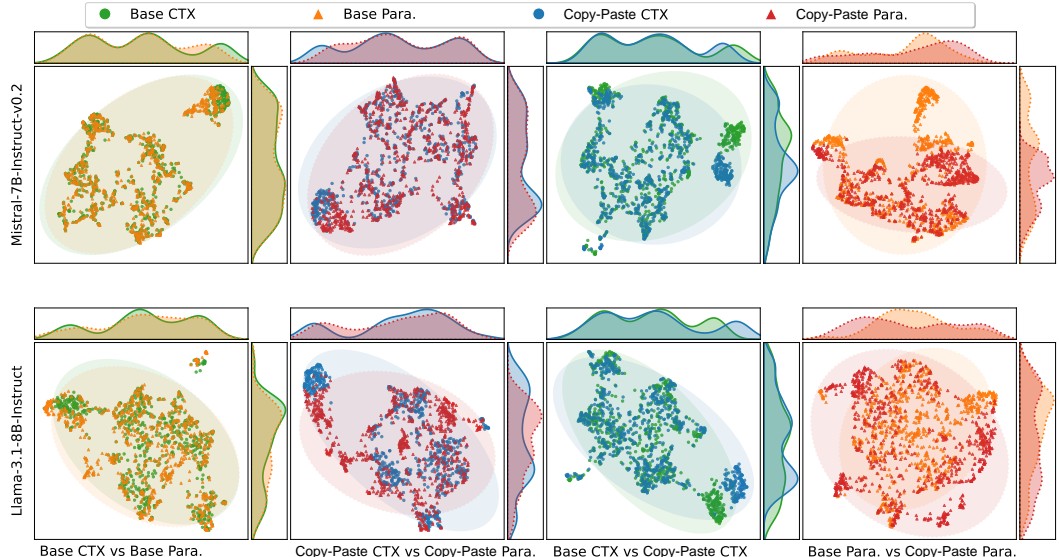

Figure 15: Dimensionality reduction visualization of hidden states distributions between contextual (CTX) and parametric (Para.) knowledge on RAGTruth dataset across two base models. Each subplot shows pairwise comparisons with marginal KDE distributions and confidence ellipses.

feed-forward networks (FFNs). Understanding how CopyPasteLLM affects attention patterns across layers and how FFNs process contextual versus parametric information could reveal finer-grained mechanisms underlying our approach's effectiveness and potentially inform more targeted architectural modifications.

**Multimodal Contextual Faithfulness:** An intriguing extension involves applying the copy-paste paradigm to multimodal scenarios, particularly in domains like medical imaging where models might favor parametric knowledge over visual evidence. For instance, when interpreting medical images, models may overlook subtle but critical visual details (such as minor variations in ECG waveforms or radiological abnormalities) in favor of common parametric patterns. Investigating whether copy-paste principles can be adapted to enforce stronger reliance on visual context—perhaps through visual attention mechanisms or multimodal copying strategies—represents a compelling avenue for enhancing faithfulness in vision-language tasks.

# L    PROMPTS

Here are the prompts we use in our experiments.

## L.1 COPY-PASTE-PROMPTING METHODS

### L.1.1 RELATED SENTENCE EXTRACTION

---
**Related Sentence Extraction**

Instruction: Please carefully read the Context and extract ALL relevant complete sentences that could help answer the Query. Output each extracted sentence on a separate line, preceded by "EXTRACTED: ".
Context
{*context*}
Query
{*query*}
CRITICAL REQUIREMENTS
1. You MUST extract complete sentences EXACTLY as they appear in the Context.
2. NO modifications, paraphrasing, or combining of sentences allowed.
3. Each extracted sentence must be highly relevant to the Query.
4. Extract ALL sentences that could help answer the Query (err on the side of inclusion).
5. Preserve all terminology, measurements, and symbols exactly as written.
Output Format
EXTRACTED: [First complete sentence exactly as it appears in Context]
EXTRACTED: [Second complete sentence exactly as it appears in Context]
...
Your extraction:

---

### L.1.2 CP-ORDER

---
**CP-Order**

Instruction: Given the Query and a list of Copied Sentences, please determine the optimal order for these sentences to create the most logical, coherent, and helpful response.
Query
{*query*}
Copied Sentences
{*numbered_sentences*}
Important Requirements
- Only use the sentence IDs provided above
- Include ALL sentences in your ordering
- Consider the query context when determining the most logical flow
Output Format
Output the optimal order as a comma-separated list of sentence IDs as below, do not provide any other information.
ORDER: [comma-separated list of sentence IDs, e.g., SENT_2,SENT_1,SENT_3]

---

### L.1.3 CP-LINK

---
**CP-Link**

Instruction: You are a professional text organization expert. Generate concise transition sentences to connect the core sentences and make the response flow naturally.
Query {*query*}
Core Sentences {*numbered_sentences*}
Requirements
1. Transition sentences should be concise (no more than 15 words)
2. They should logically connect adjacent core sentences
3. Focus on creating smooth flow between ideas
4. Common types: progression, contrast, addition, conclusion
Output Format
[TRANSITION_1_2]transition sentence content[/TRANSITION_1_2]
[TRANSITION_2_3]transition sentence content[/TRANSITION_2_3]
...
Optionally add:
[INTRO]introduction sentence[/INTRO]
[CONCLUSION]conclusion sentence[/CONCLUSION]
Please generate transitions:

---

### L.1.4 CP-REFINE

---
**Copying Requirements**

1. RELEVANT CONTEXT REUSE: Incorporate relevant text.
2. MINIMAL ORIGINAL CONTENT: Limit additions to essential connections only.
3. PRESERVE EXACT WORDING: Keep original phrases and expressions.
4. CONTEXT-ONLY INFORMATION: Use only facts explicitly in the context, do not make up any information.
5. KEEP FLUENT and NATURAL ENGLISH.

---

---

### Writer w/o Reviewer's Suggestions

Instruction: You are writer, skilled at copying relevant content from context to answer user questions. Generate highly copying responses from the given context.
Query
{*query*}
Context
{*context*}
Copying Requirements
{*copying_requirements*}
Answer:

---

### Writer w/ Reviewer's Suggestions

Instruction: You are Writer, skilled at copying relevant content from context to answer user questions. The Reviewer has suggested revisions to your old answer. Please provide a better answer to improve copying score and query relevance.
Your previous answer and Reviewer's suggestions
Old Answer
{*old_answer*}
Reviewer's Suggestions
{*reviewer_suggestions*}
Context
{*context*}
Query
{*query*}
Copying Requirements
{*copying_requirements*}
Answer:

---

### Reviewer

Your task is to review the answer to the query and suggest revisions with the goal of improving the answer's copying score (contextual faithfulness) and query relevance.
Context
{*context*}
Query
{*query*}
Answer Awaiting Review
{*answer*}
Review Criteria
- Copying Score: Text reuse from context (Current: {*copying_score*})
  - If copying score $\leq$ {*copying_threshold*}, require more context incorporation
- Contextual Faithfulness: All facts sourced from context only
  - Remove any facts or knowledge not in context
  - Reduce excessive or unnecessary original content
- Query Relevance: Direct addressing of user query
Provide CONCISE and ACTIONABLE suggestions (max 3 points):

---

## L.2 BASELINES OF PROMPT-BASED

### L.2.1 BASE

---

### Base

{*query*}

---

### L.2.2 ATTRIBUTED

---

### Attributed

Instruction: Bear in mind that your answer should be strictly based on the following context.
Context: {*context*}
Query: {*query*}
Answer:

---

### L.2.3 CITATIONS

---
**Citations**

Instruction: Bear in mind that your answer should be strictly based on the following numbered passages. Add citations in square brackets [1], [2, 3], etc. at the end of sentences that are supported by the evidence.
Numbered Sentences
{*numbered_sentences*}
Query
{*query*}
Answer:

---

## L.3 PROMPTS OF LLM JUDGES

We design the pairwise-comparison template and instructions to enable systematic, fine-grained evaluation of hallucinations in RAG responses.

---
**Pairwise Comparison Template**

Instruction: You are an expert judge. Compare two RAG responses (Response A and Response B) {*instruction*}
Context: {*context*}
Response A: {*response_a*}
Response B: {*response_b*}
Please note: Do not question or doubt the provided context. Assume the context is absolutely correct, and make your verdict strictly based on this premise.
Output Format: {{ "verdict": "<A/B/TIE>"}}

---

Above template is method-agnostic: it presents two anonymous responses, a common context treated as ground truth, and requires judges to output a formatted verdict—A, B or Tie.

The three instructions below can be slotted into the {*instruction*} placeholder in the above template and each then serves to pick the response exhibiting fewer RAG hallucinations along its respective dimension. Fabrication focuses on statements that are wholly unanchored in the provided context. Information-Distortion focuses on statements that misalign with the explicitly given context. False-Association focuses on claims that misweave separate pieces of context into an unsupported whole.

---
**Instruction for Comparing Twist Hallucination**

for information distortion hallucination. The Core Definition of Information Twist: Altering key information in the Context (e.g., numbers, timelines, subjects, conclusions).
Which has fewer information distortion hallucinations?

---

---
**Instruction for Comparing Causal Hallucination**

for causal hallucination. The Core Definition of Causal: Forcibly linking unrelated content in the Context to form new conclusions unsupported by the Context.
Which has fewer false association hallucinations?

---

## L.4 HIT RATE

---
**Hit Rate**

Context: {*context*}
Question: {*question*}
Based on the context, let's think step-by-step and answer the question in detail. Answer:

---

## L.5 ACCURACY

---
**Accuracy**

Context: {*context*}
Question: {*question*}
Options: {*options*}
Based on the above context, answer the question. You must output only a single token: A, B C or D. Do not provide any explanation or reasoning, just the chosen option. Answer:

---

## L.6 ACCURACY AFTER COT

---

**Accuracy After CoT**

Context: {*context*}
Question: {*question*}
Options: {*options*}
Based on the context, let's think step-by-step. Give your reasoining process first, then provide the final option.
Output format should be a JSON object with two fields: "reasoning" and "answer", such as: { "reasoning": "...", "answer": "..." } Answer:

---

# M    USE OF LLMS

We used large language models solely for proofreading purposes to check spelling and grammatical errors in this paper.

