# OpenReview forum: "Copy-Paste to Mitigate Large Language Model Hallucinations"
_ICLR.cc/2026/Conference — ICLR 2026 Poster_

### Official Review · Reviewer_S4zF · 2025-10-27

**Soundness:** 3
**Presentation:** 3
**Contribution:** 3
**Rating:** 4
**Confidence:** 4

**Summary:**

This paper proposes CopyPasteLLM, a two-stage framework to mitigate contextual faithfulness hallucinations in retrieval-augmented generation (RAG) systems. The authors observe an inverse correlation between copying degree and hallucination density, motivating them to develop methods that encourage high-copying behavior. The first stage generates high-copying responses through three prompting paradigms (CP-Order, CP-Link, CP-Refine), while the second stage uses these responses as preference data for Direct Preference Optimization (DPO) training. The method achieves 12.2%-24.5% improvements on FaithEval using only 365 training samples. The authors also propose a Context-Parameter Copying Capturing algorithm that reveals the model recalibrates reliance on parametric knowledge rather than enhancing contextual representations.

**Strengths:**

1. **Strong empirical results with exceptional data efficiency**: The method achieves significant improvements (12.2%-24.5% on FaithEval) using only 365 training samples, which is 50× fewer than the strongest baseline (Context-DPO with 18,000 samples).
2. **Valuable empirical observation**: The inverse correlation between copying degree and hallucination density provides an interesting insight that could inspire further research in this area.
3. **Comprehensive experimental evaluation**: The paper evaluates on multiple datasets (RAGTruth, FaithEval, PubMedQA, ConFiQA) across both counterfactual and original contexts, with multiple model architectures.
4. **Practical and implementable approach**: The automated preference data construction pipeline and straightforward copying mechanisms make the method accessible for practical deployment.
5. **Mechanistic insights**: The Context-Parameter Copying Capturing algorithm provides interpretable analysis of how the model balances contextual versus parametric knowledge during generation.

**Weaknesses:**

1. **Idealized Assumption of Infallible Context:** The framework operates under the strong assumption that the retrieved context is always correct and sufficient. In real-world applications, retrieved information can be inaccurate, biased, or incomplete. By training the model to trust the context unconditionally, this method might inadvertently discourage a crucial capability: the ability to critically assess or question a flawed context. Treating the context as absolute truth may not always be desirable for robust and reliable AI systems.
2. **Reliance on Lexical Metrics for Semantic Faithfulness:** The core metrics for copying, copy coverage ($\kappa$) and copy density ($\delta$), are based on lexical overlap. While empirically shown to be effective here, faithfulness is ultimately a semantic concept. It is possible for a response to have low lexical overlap but be a perfect, faithful paraphrase. Conversely, a response with high lexical overlap could subtly alter the meaning by changing a single critical word (e.g., "not," "only"). The reliance on a lexical proxy may not fully capture the nuances of semantic fidelity.
3. **Lack of Deeper Theoretical Grounding:** The paper presents a strong empirical correlation between high copy-degree and reduced hallucination but lacks a theoretical analysis to explain this causation more formally. Following the previous point, it would be beneficial to discuss why forcing lexical copying successfully translates into a preference for semantic faithfulness at a deeper level. How would this approach compare, for instance, to an alternative that uses a semantic similarity metric (instead of lexical overlap) as its primary optimization target during the preference generation stage (or combine lexical overlap and semantic similarity metric)?

**Questions:**

1. **Rationale for Training Data Size:** The choice of 365 training samples is quite specific. Could the authors clarify if this number has a particular significance (e.g., the size of a specific training split from a source dataset)? More importantly, have the authors investigated the performance scaling of CopyPasteLLM with respect to the number of preference pairs? It would be interesting to see if performance continues to climb with more data (e.g., 1k or 5k samples) or if it plateaus quickly, which would further underscore the method's data efficiency.
2. **Generalizability Beyond RAG:** The copy-paste mechanism is inherently tied to the presence of a source context. Could the authors elaborate on the potential generalizability of this approach beyond the classic RAG setting? For instance, could the core principle be adapted to other tasks where faithfulness to a source is paramount, such as abstractive summarization (remaining faithful to the original article) or long-form dialogue (remaining consistent with conversation history)?

---

> ### Author Response · Authors · 2025-11-25
> **Reply to Reviewer S4zF: Part 1**
>
> We are deeply grateful for your positive assessment of our work and for the insightful, thought-provoking questions regarding the theoretical underpinnings and generalizability of the CopyPasteLLM. Your comments have significantly enhanced the clarity and theoretical depth of our manuscript.
>
> ---
>
> ### Response to Q1
>
> > **Comment:** "**Idealized Assumption of Infallible Context:** The framework operates under the strong assumption that the retrieved context is always correct and sufficient. In real-world applications, retrieved information can be inaccurate, biased, or incomplete. By training the model to trust the context unconditionally, this method might inadvertently discourage a crucial capability: the ability to critically assess or question a flawed context. Treating the context as absolute truth may not always be desirable for robust and reliable AI systems."
>
> We sincerely thank you for raising this critical point. We fully agree that in real-world applications, retrieved context can be flawed, biased, or incomplete, and an ideal AI system should indeed possess the capability to critically assess the validity of its input rather than trusting it blindly.
>
> However, we respectfully submit that verifying the veracity of the source text falls outside the specific scope of this work. We focused our study on **Contextual Faithfulness**—ensuring the model adheres to the provided information—for the following reasons:
>
> **1) Responsibility of Retriever and Generator**
>
> In a RAG architecture, the responsibility for information quality typically lies with the *Retriever* or the knowledge base curator. CopyPasteLLM is designed specifically as a *Generator* component. Its primary mandate is to solve the "knowledge conflict" problem where models ignore retrieved evidence in favor of internal parametric priors.
>
> **2) Query Intent in High-Stakes Domains**
>
> Our research is motivated by scenarios where users specifically seek answers derived strictly from authoritative sources, such as clinical practice guidelines or legal statutes. In these cases, the user's intent is often: *"Tell me exactly what this document says about my problem,"* rather than asking the model to debate the document's validity. Trusting the context is a prerequisite for this specific utility.
>
> **3) CopyPaste as a Safety Attribution Mechanism**
>
> Crucially, the CopyPaste mechanism inherently mitigates the risk of incorrect context through transparency. Because our method enforces direct lexical reuse (fastly verifiable via string matching), it provides explicit attribution. This allows users—especially in specialized domains with complex terminology—to instantly trace the model's response back to the source text. If the context *is* incorrect, the user can easily identify the error because the model has quoted it directly, rather than hallucinating a plausible-sounding but fabricated correction.
>
> **4) No Impairment: CopyPasteLLM Maintains Base Model Performance**
>
> Under the assumption that the context is the "ground truth" (a standard setting in faithfulness benchmarks like FaithEval and RAGTruth), our method achieves SoTA performance, significantly outperforming baselines in resisting parametric hallucinations.
>
> In non-counterfactual settings (i.e., PubMedQA and ConFiQA with Original Context, as shown in Tables 1, 2, and 3), CopyPasteLLM still consistently improves the model's answering accuracy. This demonstrates that it does not impair the model's fundamental capabilities.
>
> | Method | PubMedQA-A | ConFiQA (QA) | ConFiQA (MR) | ConFiQA (MC) |
> | :--- | :--- | :--- | :--- | :--- |
> | Base | 88.60 | 96.22 | 71.20 | 72.27 |
> | CopyPasteLLM (Ours) | **91.40** | **97.43** | **91.87** | **91.20** |
> > **Table 1** Accuracy results on Mistral-7B-v0.2 in non-counterfactual settings. Experimental data excerpt from Table 3 of manuscript for your convience.
>
>
> | Method | PubMedQA-A | ConFiQA (QA) | ConFiQA (MR) | ConFiQA (MC) |
> | :--- | :--- | :--- | :--- | :--- |
> | Base | 97.3 | 98.02 | 93.00 | 91.02 |
> | CopyPasteLLM (Ours) | **97.5** | **99.30** | **97.17** | **96.27** |
> > **Table 2** Accuracy results on Llama-3-8B in non-counterfactual settings (PubMedQA-Artificial). Experimental data excerpt from Table 3 of manuscript for your convience.
>
>
> | Method | PubMedQA-A | ConFiQA (QA) | ConFiQA (MR) | ConFiQA (MC) |
> | :--- | :--- | :--- | :--- | :--- |
> | Base                | **98.15** | 97.93 | 89.48 | 89.97 |
> | CopyPasteLLM (Ours) | 97.67 | **99.02** | **94.95** | **94.92** |
> > **Table 3** Accuracy results on Llama-3.1-8B in non-counterfactual settings (PubMedQA-A for Artificial). Experimental data excerpt from Table 3 of manuscript for your convience.
>
> `(Continues in next part)`

---

> ### Author Response · Authors · 2025-11-25
> **Reply to Reviewer S4zF: Part 2**
>
> **5) Extended Experiment: Evaluate CopyPasteLLM's Performance across different Reasoning Difficulty**
>
> | Method       | Consensus  | Negotiation |
> | :---         | :---:      | :---:       |
> | Base         | 75.49%     | 48.91%      |
> | CopyPasteLLM | **86.85%** | **51.27%**  |
> > **Table 4** Performance breakdown on PubMedQA-Labeled by reasoning difficulty. Llama-3-8B-Instruct as Base.
>
> | Method       | Consensus  | Negotiation |
> | :---         | :---:      | :---:       |
> | Base         | 76.79%     | **46.38%**  |
> | CopyPasteLLM | **84.58%** | 44.93%      |
> > **Table 5** Performance breakdown on PubMedQA-Labeled by reasoning difficulty. Llama-3.1-8B-Instruct as Base.
>
> Furthermore, we conduct a new experiment to evaluate the reasoning capabilities of CopyPasteLLM. Specially, on PubMedQA-Labeled subset (a biomedical QA dataset where contexts contain 21% technical numeric descriptions), we split samples into:
> - "Consensus" (clear evidence in the provided context) and
> - "Negotiation" (the context may be ambiguous) depending on the experts' labeled setting from source paper of PubMedQA.
>
> As shown in above Tables 4 & 5, CopyPasteLLM yields significant gains on "Consensus" samples (+11.36% & + 8.09%) where evidence exists, while performing comparably on "Negotiation" samples (+2.36% & -1.45%). This confirms the method is effective where context is applicable, and its comparable performance to the Attributed when the context is ambiguous indicates that copy-paste strategy does not harm the model's ability to handle uncertainty.
>
> In summary, we acknowledge that enabling the generator to *also* discern context reliability is the next frontier. We have noted "Incomplete Context Scenarios" as a limitations and future work in Appendix K and agree that developing adaptive mechanisms to assess context sufficiency is a vital direction for our future work.

---

> ### Author Response · Authors · 2025-11-25
> **Reply to Reviewer S4zF: Part 3**
>
> ### Response to Q3
>
> > **Comment:** "**Lack of Deeper Theoretical Grounding:** The paper presents a strong empirical correlation between high copy-degree and reduced hallucination but lacks a theoretical analysis to explain this causation more formally. Following the previous point, it would be beneficial to discuss why forcing lexical copying successfully translates into a preference for semantic faithfulness at a deeper level. How would this approach compare, for instance, to an alternative that uses a semantic similarity metric (instead of lexical overlap) as its primary optimization target during the preference generation stage (or combine lexical overlap and semantic similarity metric)?"
>
> Thank you for giving us the opportunity to clarify the mechanistic of CopyPaste effectiveness. We agree that explaining why a surface-level constraint (high-copying) translates into model-level belief (semantic faithfulness) is crucial.
>
> To address your concern, we provide a mechanistic interpretation explaining how the external constraint of high-copying responses translates into improved contextual faithfulness and reduced hallucinations within the LLM's internal dynamics. This interpretation connects the CopyPaste objective to fundamental model components, which we analyze from two perspectives: (1) Attention Dynamics and Contextual Anchoring, and (2) Information Entropy Reduction.
>
> **1)Attention Dynamics and Contextual Anchoring**
>
> In Transformer-based architectures, the probability of generating the next token $y_t$ is governed by the attention mechanism. Let the input sequence be the concatenation of the context $\mathcal{C}$ and the generated prefix $y_{<t}$. The attention output $\mathbf{h}_t$ at step $t$ is a weighted sum of value vectors:
>
> $$
>     \mathbf{h} _ t = \underbrace{\sum _ {j \in \mathcal{C}} \alpha _ {t,j} \mathbf{v} _ j} _ {\text{Contextual Attention}} + \underbrace{\sum _ {k \in y _ {<t}} \alpha _ {t,k} \mathbf{v} _ k} _ {\text{Parametric Attention}}
> $$
>
> where $\alpha_{t, \cdot}$ represents the softmax-normalized attention weights. Hallucinations typically occur when the model fails to attend to the context (low $\sum_{j \in \mathcal{C}} \alpha_{t,j}$) and instead relies on internal parametric priors activated by the generated history.
>
> **The Anchoring Effect:** We posit that CopyPaste leverages the `Induction Head` mechanism[1,2], a circuit responsible for in-context copying.
> If we enforce the previous token $y_{t-1}$ to be a direct copy of a token $c_k \in \mathcal{C}$ (where $c_k$ is the token at position $k$ in the context), the query vector $\mathbf{q} _ t$ (derived from $y_{t-1}$) will strongly correlate with the key vector $\mathbf{k}_{c_k}$ in the context.
>
> Mathematically, maximizing the copying likelihood in Stage 1 (Section 4.1.1 in manuscript) effectively optimizes the attention weights such that:
>
> $$
> \text{score}(\mathbf{q} _ t, \mathbf{k} _ {c_k}) \propto \mathbf{q} _ t^\top \mathbf{k} _ {c _ k} \gg \mathbf{q} _ t^\top \mathbf{k} _ {\text{para}}
> $$
>
> This creates a `Semantic Anchor`, forcing the attention distribution $\alpha$ to collapse onto the context:
>
> $$
> \lim _ {\text{copying} \to \text{max}} \sum _ {j \in \mathcal{C}} \alpha _ {t,j} \approx 1
> $$
>
> By ensuring $y_{t-1}$ is a copy, we mechanically guide the induction heads to retrieve the subsequent ground-truth token $c_{k+1}$ from $\mathcal{C}$, thereby physically suppressing the attention pathways that lead to parametric hallucinations. This aligns with our empirical observation in Figure 3 of manuscript, where CopyPasteLLM exhibits significantly suppressed parametric logits.
>
> `(Continues in next part)`

---

> ### Author Response · Authors · 2025-11-25
> **Reply to Reviewer S4zF: Part 4**
>
> **2) Entropy Reduction and Search Space Constriction**
>
> From an information-theoretic perspective, faithfulness hallucinations arise from high uncertainty in the conditional distribution $P(y _ t | \mathcal{C}, y _ {<t})$. Let $\mathcal{V}$ be the full vocabulary of the LLM, and $\mathcal{V}_{\mathcal{C}} \subset \mathcal{V}$ be the subset of tokens present in the context.
>
> The CopyPaste objective imposes a constraint that the generated response $Y$ must maximize lexical overlap with $\mathcal{C}$. This effectively serves as a regularization term that constrains the search space $\Omega$ from the vast $\mathcal{V}$ to the much smaller $\mathcal{V}_{\mathcal{C}}$.
>
> The conditional entropy of the generation step without constraints is:
>
> $$
>     H(Y|\mathcal{C}) _ {\text{Base}} = - \sum_{w \in \mathcal{V}} P_{\text{Base}}(w|\mathcal{C}) \log P_{\text{Base}}(w|\mathcal{C})
> $$
>
> In standard generation, the probability mass is often distributed over a long tail of semantically similar but extrinsically hallucinated tokens from parametric memory.
> In contrast, CopyPasteLLM is optimized to concentrate probability mass on $\mathcal{V}_{\mathcal{C}}$:
>
> $$
>     \sum _ {w \in \mathcal{V} _ {\mathcal{C}}} P _ {\text{CP}}(w|\mathcal{C}) \to 1 \implies P _ {\text{CP}}(w \notin \mathcal{V} _ {\mathcal{C}}) \to 0
> $$
>
> Consequently, the entropy of the CopyPaste distribution is strictly lower than that of the base distribution:
>
> $$
>     H(Y|\mathcal{C}) _ {\text{CP}} \ll H(Y|\mathcal{C}) _ {\text{Base}}
> $$
>
> By minimizing the entropy and pruning the probability of tokens $w \notin \mathcal{V}_{\mathcal{C}}$, we statistically minimize the risk of sampling hallucinated content. This theoretical result explains why our method, despite relying on lexical proxies, effectively improves semantic faithfulness by eliminating the `lexical pathways` that allow parametric priors to leak into the generation.
>
> We believe this mechanistic interpretation provides the necessary theoretical grounding to explain the strong empirical results observed in the manuscript. Above content has been added to Appendix A of manuscript.
>
> > [1] Olsson, Catherine, et al. "In-context learning and induction heads." arXiv preprint arXiv:2209.11895 (2022).
> > [2] Huang, Lei, et al. "Improving contextual faithfulness of large language models via retrieval heads-induced optimization." arXiv preprint arXiv:2501.13573 (2025).

---

> ### Author Response · Authors · 2025-11-25
> **Reply to Reviewer S4zF: Part 5**
>
> ### Response to Q2
>
> > **Comment:** "**Reliance on Lexical Metrics for Semantic Faithfulness:** The core metrics for copying, copy coverage ($\kappa$) and copy density ($\delta$), are based on lexical overlap. While empirically shown to be effective here, faithfulness is ultimately a semantic concept. It is possible for a response to have low lexical overlap but be a perfect, faithful paraphrase. Conversely, a response with high lexical overlap could subtly alter the meaning by changing a single critical word (e.g., "not," "only"). The reliance on a lexical proxy may not fully capture the nuances of semantic fidelity."
>
> Thanks for this nuanced observation. We fully agree that faithfulness is ultimately a semantic concept and that high lexical overlap does not intrinsically guarantee semantic fidelity (e.g., the omission of a critical "not").
>
> However, we advocate for lexical copying not as a *replacement* for semantic evaluation, but as a training constraint to suppress contextual faithfulness hallucinations (as stated in the reponse to Q3 above). Here, We clarify this concern through three aspects:
>
> **1) Paraphrasing as a Hallucination Risk**
>
> As stated in the reponse to Q3 above, while faithful paraphrasing is theoretically possible, our mechanistic interpretation suggests it is operationally risky for RAG systems. To paraphrase, an LLM must access its internal vocabulary distribution ($\mathcal{V}$). Maybe shifting probability mass to the parametric "long tail" activates the same attention pathways where hallucinations reside. By enforcing lexical copying, we mechanically constrain the search space to the context ($\mathcal{V}_{\mathcal{C}}$). This acts as a regularizer that forces the model to copying evidence rather than interpret it, aim to reduce the entropy that leads to hallucination.
>
> **2) Semantic Filters Guard Against High-Copying Mistakes**
>
> We explicitly address this in our pipeline design (Multi-Criteria Filtering in Stage 2). We do *not* select training data based on copying metrics alone. All high-copying candidates must pass a "semantic check" using faithfulness metrics ( AlignScore and MiniCheck). If a generated response has high lexical overlap ($\kappa$) but changes the meaning (e.g., by dropping a negation), the semantic faithfulness scores will drop. These candidates are filtered out before DPO training, ensuring the model only learns from responses that are both lexically grounded and semantically faithful.
>
> **3) Assessing CopyPaste's Contextual Semantic Understanding**
>
>
>
> | CI Level | ConFiQA-QA | ConFiQA-MR | ConFiQA-MC |
> | :---: | :---: | :---: | :---: |
> | 95% | 63.57 (62.74 - 64.39) | 81.56 (80.68 - 82.45) | 84.59 (83.76 - 85.43) |
> | 99% | 63.57 (62.35 - 64.79) | 81.56 (80.25 - 82.87) | 84.59 (83.38 - 85.80) |
> > **Table 6** CopyPasteLLM Accuracy after CoT Comparison Across ConFiQA (unseen) Subsets (with 95% and 99% CIs). The base model performance are 39.7, 46.7 and 24.6 across QA, MR and MC subset.
>
> While our training objective relies on lexical proxies, our evaluation uses strict accuracy metric. We conduct a new experiment that request CopyPasteLLM to generate the reasoning preocess firstly, and then output the final verdict (where the model's output is constrained to the format {"reasoning": "...", "answer": "..."}). We evaluate the final verdict if include ground answers. In other words, the accuracy represent whether the model can understand the semantic of context. As shown in Table 6, CopyPasteLLM gain +23.87%, +34.86% and +59.99% accuracy across ConFiQA subsets.
>
> In summary, while lexical overlap is a proxy, it is a highly effective operational proxy. It forces the model to anchor its attention on the source text, while our semantic filtering pipeline ensures that this copying does not come at the cost of accuracy.

---

> ### Author Response · Authors · 2025-11-25
> **Reply to Reviewer S4zF: Part 6**
>
> ### Response to Q4
>
> > **Comment:** "**Rationale for Training Data Size:** The choice of 365 training samples is quite specific. Could the authors clarify if this number has a particular significance (e.g., the size of a specific training split from a source dataset)? More importantly, have the authors investigated the performance scaling of CopyPasteLLM with respect to the number of preference pairs? It would be interesting to see if performance continues to climb with more data (e.g., 1k or 5k samples) or if it plateaus quickly, which would further underscore the method's data efficiency."
>
> Thanks for these insightful questions.
>
> **1) About `Could the authors clarify if this number has a particular significance?`**
>
> We must clarify that the 365 training samples were not chosen arbitrarily. Instead, they were carefully filtered as high-quality preference pairs through our designed, fully automated Stage 2 pipeline. This rigorous selection process yielded a diverse set of samples encompassing various domains, such as science (FaithEval), biomedicine (PubMedQA), and daily life (RAGTruth). Consequently, this final collection of 365 samples represents the outcome of this automated filtering methodology, and the number "365" itself holds no special significance.
>
> **2) About `More importantly, have the authors investigated the performance scaling of CopyPasteLLM with respect to the number of preference pairs?`**
>
> This is a very instereting question! We conduct a new experiment to address your questions.
>
> We intentionally limited ourselves to this small scale to test the extreme data efficiency of our approach, contrasting it with baselines that typically require 10k+ samples (e.g., Context-DPO uses 18k, ParamMute uses 32k).
>
> |   Step | ConFiQA-QA | ConFiQA-MR | ConFiQA-MC |
> |-------:|------:|------:|------:|
> |     10 | 42.01 | 51.38 | 33.51 |
> |     20 | 45.87 | 61.19 | 49.47 |
> |     30 | 47.87 | 67.84 | 62.30 |
> |     40 | 59.54 | 80.78 | 84.26 |
> |     50 | **61.79** | **84.02** | **88.49** |
> |     60 | 60.25 | 81.91 | 86.29 |
> |     70 | 61.81 | 81.68 | 86.15 |
> |     80 | 62.64 | 82.03 | 86.28 |
> |     90 | 62.53 | 82.81 | 86.59 |
> |    100 | 64.62 | 84.01 | 88.23 |
> |    110 | 65.79 | **84.5**  | 87.81 |
> |    120 | **67.10** | **85.25** | 88.10 |
> |    130 | 66.74 | 84.51 | 87.74 |
> |    140 | 64.79 | 82.89 | 86.37 |
> |    150 | 64.10 | 82.26 | 85.03 |
> |    160 | 63.46 | 82.06 | 84.54 |
> |    170 | 63.34 | 82.24 | 84.56 |
> |    180 | 63.52 | 81.8  | 84.03 |
> |    190 | 63.21 | 81.78 | 84.23 |
> |    200 | 63.47 | 81.99 | 84.35 |
> |    210 | 63.68 | 82.09 | 83.97 |
> |    218 | 63.57 | 81.56 | 84.59 |
> > **Table 7** CopyPasteLLM Accuracy across Training Size. We use bold font to highlight each peak. We also draw the curves at Figure 12 in revised manuscript.
>
> As shown in Table 7, CopyPasteLLM exhibits extremely rapid convergence. The performance curves rise sharply and reach a plateau around step 110–120 (approx. 1 epoch). Notably, on complex reasoning tasks like ConFiQA-MR and ConFiQA-MC, the model approaches peak performance as early as step 50. This rapid saturation indicates that the "high-copying preference signal" constructed by our pipeline is highly potent. The model realigns its internal belief mechanism with minimal updates. Consequently, the marginal utility of increasing data volume diminishes quickly. Hence, it suggests that the performance gains of CopyPasteLLM are driven primarily by the quality of the preference pairs rather than the sheer scale of the dataset.
>
> In summary, our results indicate that CopyPasteLLM does indeed "plateau quickly" in a positive sense—it learns the desired behavior with very few samples, underscoring the method's exceptional data efficiency. Future work may explore scaling to 1k or 5k samples to larger base model for broader domain adaptation, but our current findings suggest the core mechanism of contextual faithfulness can be established with small data.

---

> ### Author Response · Authors · 2025-11-25
> **Reply to Reviewer S4zF: Part 7**
>
> ### Response to Q5
>
> > **Comment:** "**Generalizability Beyond RAG:** The copy-paste mechanism is inherently tied to the presence of a source context. Could the authors elaborate on the potential generalizability of this approach beyond the classic RAG setting? For instance, could the core principle be adapted to other tasks where faithfulness to a source is paramount, such as abstractive summarization (remaining faithful to the original article) or long-form dialogue (remaining consistent with conversation history)?"
>
> Thanks for this forward-looking question! We agree that the core principle of CopyPaste—using lexical constraints to mechanically suppress parametric priors—extends well beyond standard RAG. Whenever a task requires grounding generation in specific input data (source text, conversation history, or even other modalities) rather than the model's internal memory, we believe the CopyPaste mechanism is applicable.
>
> **1) About `such as abstractive summarization (remaining faithful to the original article`**
>
> In this context, hallucinations often manifests as the generation of entities or events not present in the source article.
>
> As discussed in response of Q3, CopyPaste works by constraining the search space $\Omega$ to the vocabulary of the context ($\mathcal{V}_{\mathcal{C}}$). For summarization, we can still adapt CP-Refine method to reward summaries that maximize the reuse of distinct entities and key phrases from the source article, effectively treating the article as the "context." Ensuring the summary remains faithful to the source article's factual content while allowing for structural abstraction.
>
> **2) About `or long-form dialogue (remaining consistent with conversation history)?`**
>
> In long-form dialogue, contextual faithfulness translates to consistency with conversation history. Models often hallucinate user details or contradict previous turns because they revert to generic parametric priors instead of attending to the specific history. CopyPaste creates a "Semantic Anchor" that collapses attention onto the provided input. In dialogue, we can treat the conversation history as the Context $\mathcal{C}$. For example, by training the model to copy constraints or facts directly from the conversation history (e.g., "As you mentioned earlier... [copyied_fragemtns]"), we can force the induction heads to retrieve specific historical tokens, thereby maintaining consistency and reducing persona drift.
>
> **3) Extension to Multimodal Contexts**
>
> We have explicitly discussed extending this principle to multimodal settings in our manuscript. In domains like medical imaging, models often favor parametric knowledge over visual evidence (e.g., ignoring a specific anomaly in an image because it is rare). The core principle of CopyPaste can be conceptualized here as enforcing "Visual Anchoring"—mechanistically forcing the model to attend to specific image patches (visual context) before generating a diagnosis, rather than relying on the statistical likelihood of diseases (parametric prior).
>
> In summary, the CopyPaste philosophy is not limited to RAG; it is a potentially generalizable strategy for any task where extrinsic evidence must take precedence over intrinsic priors.

---

### Official Review · Reviewer_ujdf · 2025-11-01

**Soundness:** 3
**Presentation:** 3
**Contribution:** 3
**Rating:** 6
**Confidence:** 3

**Summary:**

This paper proposes CopyPasteLLM, a two stage framework to reduce context unfaithful hallucination in RAG. In stage 1 they create high copying answers via three prompting models, CP-Order, CP-link, CP-Refine, while in the stage 2 they convert these candidates plus standard baselines into preference data using multi criteria filtering.
Copying degree is quantified with Copy Coverage and Copy Density, adapted from Newsroom. The paper also introduces Context Parameter Copying Capturing, a token-level probe that compares with-context vs no-context decoding to attribute reliance to contextual versus parametric knowledge through top-k logits along chain of thought. Experiments on various datasets report large gains with only 365 samples for DPO.

**Strengths:**

1) The author clearly operationalize “copying” as a proxy for faithfulness by using κ, δ metrics which are defined cleanly and tied to an algorithmic fragment detector, the paper keeps the goal explicit: increase lexical reuse from context while balancing relevance and fluency.
2) The proposed CP-Order, CP-Link, and CP-Refine span hard to soft constraints, making the Stage 1 data diverse yet biased toward high copying.
3) The authors also show that how by using with just 365 pairs for DPO, CopyPasteLLM outperforms Context-DPO trained on 18k pairs and other fine-tuning baselines across counterfactual settings, with large margins on FaithEval; non-counterfactual accuracy also improves

**Weaknesses:**

1) Lexical copying is not faithfulness, κ, δ reward verbatim reuse and long spans and a response can copy irrelevant or misleading context sentences which yet still be unfaithful to the question. The paper partially addresses relevance via embeddings and perplexity, but κ, δ might elevate answers that are faithful to the wrong passage.
2) Stage 1 filtering and Stage 2 tournament rely on AlignScore, MiniCheck, and a Qwen-32B judge, plus embedding similarity for relevance. These components are themselves models with biases and variance, having inter annotator agreement with human would be essential.
3) The “stamping” step may entangle selection by copying degree with label assignment. Without ablations, it’s hard to tell whether gains come from copying behavior or from aggressive negative construction.

**Questions:**

1) Table 3 states PubMedQA is evaluated on 20k samples, whereas in Table 4 it is mentioned size of PubMedQA as 1000, this needs some clarification.
2) Authors should introduce some sort of confidence intervals to their results reported, as the current results are point wise, and CI should strengthen their claims, also across multiple seed values.

---

> ### Author Response · Authors · 2025-11-25
> **Reply to Reviewer ujdf: Part 1**
>
> We sincerely thank you for the thoughtful and precise critique, especially for pointing out the need for disentangled ablations and confidence-interval analyses, which directly guided the expanded experiments in our revision. We also appreciate you recognizing the contribution and data efficiency of the CopyPasteLLM framework.
>
> ---
>
> ### Response to Q1
>
> > **Comment:** "Lexical copying is not faithfulness, $\kappa, \delta$ reward verbatim reuse and long spans and a response can copy irrelevant or misleading context sentences which yet still be unfaithful to the question. The paper partially addresses relevance via embeddings and perplexity, but $\kappa, \delta$ might elevate answers that are faithful to the wrong passage."
>
> We fully agree with you that lexical copying metrics ($\kappa$ and $\delta$) are not responsible for query relevancy. To address your concerns, we will first briefly clarify how our method ensures that the copied fragments remains relevant to the query during preference pair construction, and then conduct 2 new experiment to demonstrate that the qeury relevance of the copied fragments is superior to or comparable with the baselines.
>
> **1) How Our Method Ensures Query Relevancy in Responses**
>
> In Stage 1 of our pipeline, the high-copying preference pairs used to train CopyPasteLLM are derived from three Prompting-Based methods (CP-Order, CP-Link, and CP-Refine).
>
> - For CP-Order and CP-Link, the sentences to be copied are first obtained by an Extractor Agent using the instruction: *"Please carefully read the Context and extract all relevant complete sentences that could help answer the Query..."*
> - CP-Refine employs a two-agent framework (Writer & Reviewer). The copied fragments are determined by the Writer and are supervised by the Reviewer, whose instruction is: *"Your task is to review the answer to the query and suggest revisions with the goal of improving the answer’s copying score (contextual faithfulness) and query relevance..."*
>
> Based on our observations, although the agents perform well in selecting query-relevant sentences, we further enhance this process in Stage 2. We use the Qwen3-Embedding-8B text embedding model to calculate the cosine similarity between the response and the query, filtering out responses with low query relevance. These filtered responses are then used as rejected samples in DPO training, thereby ensuring that CopyPasteLLM learns to prefer high query relevance.
>
> **2) Extended Experiment 1: Comparative Query Relevancy of Copied Fragments Across Different Lengths**
>
> | Copied Fragment Length | Attributed | CoCoLex | Canoe | Context-DPO | CopyPasteLLM |
> | :--- | :---: | :---: | :---: | :---: | :---: |
> | 22 | 0.5060 | 0.5558 | 0.5473 | 0.5098 | **0.5568** |
> | 26 | 0.5451 | 0.5233 | 0.4647 | 0.5291 | **0.5598** |
> | 41 | 0.5056 | 0.6016 | 0.5211 | 0.5702 | **0.6471** |
> | 47 | 0.4603 | 0.4794 | - | 0.4918 | **0.6074** |
> | 48 | 0.5977 | 0.5708 | 0.5793 | - | **0.6670** |
> | 56 | - | 0.6389 | - | - | **0.6517** |
> | 60 | - | 0.5831 | - | - | **0.6164** |
> | 66 | - | 0.5994 | 0.5961 | 0.5579 | **0.8143** |
> | 73 | - | 0.5058 | - | - | **0.5727** |
> | 81 | - | 0.6067 | - | - | **0.7437** |
> | 87 | - | 0.5100 | - | - | **0.6918** |
> | 97 | - | 0.6075 | - | - | **0.6600** |
> | 116 | - | 0.6055 | - | - | **0.6755** |
> > **Table 1** Query Relevancy Analysis of Copied Fragments on FaithEval. Measured by `Qwen3-Embedding-8B`. Note: The full experimental range, spanning fragment lengths from 2 to 116, is detailed in Figure 11 of the manuscript.
>
>
> | Copied Fragment Length | Attributed | CoCoLex | Canoe | Context-DPO | CopyPasteLLM |
> | :--- | :---: | :---: | :---: | :---: | :---: |
> | 23 | 0.5663 | 0.5557 | 0.5857 | 0.5502 | **0.5910** |
> | 32 | 0.5899 | 0.5976 | 0.5539 | 0.5648 | **0.6005** |
> | 47 | 0.5018 | 0.5929 | - | 0.5310 | **0.6372** |
> | 61 | - | 0.5318 | 0.6255 | - | **0.6378** |
> | 66 | - | 0.6542 | 0.7001 | 0.5938 | **0.7941** |
> | 72 | - | 0.6576 | - | - | **0.6992** |
> | 75 | - | 0.6167 | - | - | **0.6366** |
> | 81 | - | 0.6960 | - | - | **0.7520** |
> | 82 | - | 0.6535 | - | - | **0.6679** |
> | 87 | - | 0.5530 | - | - | **0.7061** |
> | 97 | - | 0.6871 | - | - | **0.7474** |
> | 112 | - | 0.6123 | - | - | **0.6530** |
> > **Table 2** Query Relevancy Analysis of Copied Fragments on FaithEval. Measured by `BGE-M3`.
>
> As shown in Table 1 and 2, the fragments copied by CopyPasteLLM maintain higher cosine similarity with the query than baselines. This verifies that the model learns to selectively copy evidential fragments, rather than dumping random, irrelevant context.
>
> `(Continues in next part)`

---

> ### Author Response · Authors · 2025-11-25
> **Reply to Reviewer ujdf: Part 2**
>
> **3) Extended Experiment 2: Analysis of Response Length Combined Hit Rate**
>
> | Method                  | Length          | $\kappa$ (Coverage) | $\delta$ (Density) |
> |-------------------------|-----------------|---------------------|---------------------|
> | Attributed              | $199.0 \pm 68.29$ | $0.552 \pm 0.129$   | $2.70  \pm 2.85$    |
> | CoCoLex                 | $75.0  \pm 40.01$ | $0.989 \pm 0.040$   | $50.08 \pm 37.68$   |
> | Canoe                   | $198.0 \pm 69.30$ | $0.551 \pm 0.126$   | $2.67  \pm 3.68$    |
> | Context-DPO             | $159.5 \pm 62.72$ | $0.631 \pm 0.130$   | $4.17  \pm 4.32$    |
> | ParamMute               | $4.0   \pm 18.69$ | $1.000 \pm 0.182$   | $3.00  \pm 13.23$   |
> | CopyPasteLLM (Ours)     | $126.0 \pm 78.71$ | $0.844 \pm 0.135$   | $10.49 \pm 15.49$   |
> > **Table 3** Response Analysis by Method Under Chain-of-Thought (CoT) Instruction. The models were instructed to provide the reasoning process; hence, the reported response lengths should not too short. All metrics are reported as Median $\pm$ Standard Deviation. Experiments are based on the LLaMA-3-8B-Instruct model.
>
> | Method              | ConFiQA-QA  | ConFiQA-MR  | ConFiQA-MC  | FaithEval |
> | :---                | :---:       | :---:       | :---:       | :---: |
> | Attributed          | 91.4        | 71.5        | 53.6        | $\underline{34.2}$ |
> | CoCoLex             | 37.4        | 14.8        | 15.5        | 17.9 |
> | Canoe               | $\underline{93.2}$ | **83.8**    | $\underline{73.7}$ | 34.0 |
> | ParamMute           | 82.2        | 72.4        | 70.2        | 22.5        |
> | CopyPasteLLM (Ours) | **96.7**    | $\underline{83.4}$ | **75.9**    | **37.2** |
> > **Table 4** Performance Comparision of Hit. For convenience, the experimental data are excerpted from Table 1 in the manuscript.
>
> Additionally, our primary results (as shown in Table 3) are evaluated based on the hit rate against gold-truth answers when instructed by Chain of Thought (CoT). CopyPasteLLM demonstrates effective improvements in the hit rate while maintaining an appropriate response length. The median response length is 126 words—significantly shorter than Attributed (199 words) and Context-DPO (159.5 words), yet substantially longer than ParamMute (4 words)—as detailed in Table 4. This demonstrates that the high-copying behavior is effective, resulting in accurate query-relevant reasoning.

---

> ### Author Response · Authors · 2025-11-25
> **Reply to Reviewer ujdf: Part 3**
>
> ### Response to Q2
>
> > **Comment:** "Stage 1 filtering and Stage 2 tournament rely on AlignScore, MiniCheck, and a Qwen-32B judge, plus embedding similarity for relevance. These components are themselves models with biases and variance, having inter annotator agreement with human would be essential."
>
> We sincerely appreciate your emphasis on the importance of data quality control. We fully acknowledge that automated metrics inevitably exhibit inherent biases and variance when benchmarked against human annotation. To address this concern, we conducted a focused human-metric consistency study.
>
> Since the complexity of pairwise comparison grows quadratically with the number of candidate responses (five candidates yield 10 pairs per sample), we limited human annotation to a controlled but representative subset. We sampled 10 instances each from RAGTruth, PubMedQA, and FaithEval (30 total samples) using a fixed random seed for reproducibility. For each sample, annotators compared the responses of the five candidate methods used in our pipeline: Attributed, Citations, CP-Order, CP-Link, and CP-Refine.
>
> Three trained graduate annotators independently performed pairwise comparisons (10 pairs per sample) along the following three dimensions:
> - Contextual Faithfulness: Which response is more faithful to the context? (Comparable to the average of AlignScore and MiniCheck).
> - Hallucinations: Which response contains fewer hallucinations (e.g., rewriting or causal)? (Comparable to the hallucinations judge by Qwen3-32B).
> - Query Relevance: Which response is more relevant to the query? (Comparable to Qwen3-Embedding-8B).
>
> Annotators selected A / B / Tie, resulting in 900 total judgments per annotator (30 samples $\times$ 10 pairs $\times$ 3 dimensions). With each sample judged 30 times, we utilized ELO scores to rank the candidates for each sample. We then computed the Spearman rank correlation ($\tau$) between the human-derived ELO ranking and the automatic metric rankings across the 30 samples.
>
> | Comparison | Spearman ( $\tau$ ) |
> | :---: | :---: |
> | Faithfulness | 0.532 |
> | Hallucinations | 0.483 |
> | Relevance | 0.515 |
> > **Table 5** Human Evaluation.
>
> The results show a moderate, reliable correlation between our automated metrics and human judgment across the core dimensions. With Spearman rank correlations ($\tau$) consistently around 0.5, specifically 0.532 for Faithfulness, 0.483 for Hallucination, and 0.515 for Relevance, we validate that the employed metrics (AlignScore, MiniCheck, Qwen3-32B and Qwen3-Embedding-8B) serve as a useful and efficient proxy for approximating human preference rankings in our filtering and tournament stages.
>
> Crucially, we strategically selected these cost-effective and relatively "cheap" automated metrics to highlight the generalizability and scalability of the CopyPaste. We emphasize that if required (e.g., for deployment in high-stakes domains), these metrics could be readily replaced by more powerful and expensive LLMs (such as Gemini 3 or GPT-5). We are confident that such a substitution would further enhance CopyPasteLLM's performance, even though its current results already surpass the majority of existing baselines.

---

> ### Author Response · Authors · 2025-11-25
> **Reply to Reviewer ujdf: Part 4**
>
> ### Response to Q3
>
> > **Comment:** "The “stamping” step may entangle selection by copying degree with label assignment. Without ablations, it’s hard to tell whether gains come from copying behavior or from aggressive negative construction."
>
> Thanks for giving us the opportunity to clarify the independent contributions of our core components! We performed a ablation study to disentangle the effects of "Copying behavior" and "Stamping Answers. The results confirm that while Copying is the core driver of faithful reasoning, Stamping acts as a necessary bridge to formalize that reasoning into a valid answer.
>
> **1) Ablation Study of Copying and Stamping**
>
> | Model        | QA/Acc.  | QA/Hit   | MR/Acc.  | MR/Hit   | MC/Acc.  | MC/Hit   |
> |----          |----      |--------  |-------   |--------  |-------   |------    |
> | Base Model   | 39.7     | 91.6     | 46.7     | 67.8     | 24.6     | 51.0     |
> | CopyPasteLLM | **63.6** | **95.0** | **81.6** | **83.4** | **84.6** | **80.1** |
> | w/o Copying  | 56.8     | 92.7     | 69.9     | 72.5     | 63.6     | 62.8     |
> | w/o Stamping | 41.9     | 93.5     | 54.8     | 73.3     | 38.4     | 62.2     |
> > **Table 6** Presents the results of an ablation study performed on the CopyPasteLLM framework using the Llama-3.1-8B-Instruct model on the ConFiQA benchmark. This study investigates the contribution of the key components (Copying and Stamping) to the overall performance. `Acc.` denotes Accuracy after Chain-of-Thought (CoT) prompting, where the model's output is constrained to the format {"reasoning": "...", "answer": "..."}. Experimental data excerpt from Figure 12 of manuscript.
>
> **Ablation Setting**
> - For `w/o Copying`: In this setting, we excluded the three CopyPaste-Prompting methods (CP-Order, CP-Link, CP-Refine) and constructed preference pairs solely from mainstream baselines (Base, Attributed, Citations). We then selected the top 365 samples based on multi-criteria filtering.
> - For `w/o Stamping`: We evaluated the role of the Stamping Answers step by removing it directly. In this variant, the model learns from the chosen response's reasoning trace without the explicit appending of the ground-truth answer label.
>
> **Experimental analysis**
>
> First, as shown in Figure 6, CopyPasteLLM consistently and significantly outperforms the `w/o Copying` variant. Even when the responses from `w/o Copying` are filtered for faithfulness, they fail to achieve the same level of reliability. This proves that the explicit copying behavior introduced by our method is the core mechanism that suppresses contexutal faithfulness hallucinations.
>
> Second, the comparison between `w/o Stamping` and CopyPasteLLM reveals the specific role of stamping.
> - On Hit Metrics: The performance of `w/o Stamping` is almost the same as `w/o Copying`.
> - On the Accuracy after CoT: However, without Stamping, the Accuracy after CoT drops drastically. We believe this divergence demonstrates that Stamping serves a necessary bridge role. It forces the model to output final verdict derived from its reasoning trace. In other words, copying bahavior responsible for providing grounding evidences in reasoning trace, and stamping it then responsible for transfering the reasoning trace to final verdict.
>
> In summary, Copying and Stamping are distinct but synergistic components. Copying ensures the model knows the faithful answer by anchoring its response to context, while Stamping ensures the model commits to that answer in the final verdict. Both are essential for the superior performance of CopyPasteLLM.

---

> ### Author Response · Authors · 2025-11-25
> **Reply to Reviewer ujdf: Part 5**
>
> ### Response to Q4
>
> > **Comment:** "Table 3 states PubMedQA is evaluated on 20k samples, whereas in Table 4 it is mentioned size of PubMedQA as 1000, this needs some clarification."
>
> We sincerely thank you for spotting this confusion regarding the PubMedQA dataset size. We apologize for the lack of clarity in the initial submission where the distinction regarding the dataset splits was not clearly stated in Table 4.
>
> The Table 4 of revised manuscript reports the size of PubMedQA using the Expert-Labeled subset (1,000 samples). This specific subset was used for detailed analysis (RQ1/RQ3) and served as the source for preference pairs in our training pipeline (RQ2).
>
> The Table 3 of revised manuscript reports the evaluation on the larger artificial subset of PubMedQA (20,000 samples, random selection is made from 211k with a fixed seed 42.) to robustly test the model's generalization capabilities on a broader range of unseen data.
>
> We have explicitly clarified this distinction in the revised Table 4 of the manuscript. We paste it to belowe for your convenience:
>
> | Dataset | Subset | Domain | Size | Gold Answer | RQ1 | RQ2 | RQ3 |
> | :--- | :--- | :--- | :--- | :--- | :--- | :--- | :--- |
> | RAGTruth | QA | Daily-Life | 839 | $\times$ | Eval | only Train (16) | Eval |
> | FaithEval | Counterfactual | Science | 1,000 | $\checkmark$ | Eval | Train / Eval (241 / 759) | Eval |
> | PubMedQA | Labeled | Biomedicine | 1,000 | $\checkmark$ | Eval | Train / Eval (108 / 892) | Eval |
> | PubMedQA | Artificial | Biomedicine | 20,000 | $\checkmark$ | - | Eval | - |
> | ConFiQA | Counterfactual & Original | Wikidata | 36,000 | $\checkmark$ | - | Eval | - |
> > **Table 4 of revised manuscript** Datasets and their roles across 3 research questions. `Train` refers to the number of samples utilized for training our CopyPasteLLM, and `Eval` refers to the number of samples used for evaluation. The 20,000 samples of the PubMedQA Artificial subset were randomly sampled using the random seed 42 from the 211k entries.

---

> ### Author Response · Authors · 2025-11-25
> **Reply to Reviewer ujdf: Part 6**
>
> ### Response to Q5
>
> > **Comment:** "Authors should introduce some sort of confidence intervals to their results reported, as the current results are point wise, and CI should strengthen their claims, also across multiple seed values."
>
> Thanks for this constructive suggestion to include confidence intervals, which is crucial for establishing the statistical robustness of our claims! We have since conducted new, extensive experiments to compute these intervals.
>
> | CI Level | QA | MR | MC |
> | :---: | :---: | :---: | :---: |
> | 95% | 63.57 (62.74 - 64.39) | 81.56 (80.68 - 82.45) | 84.59 (83.76 - 85.43) |
> | 99% | 63.57 (62.35 - 64.79) | 81.56 (80.25 - 82.87) | 84.59 (83.38 - 85.80) |
> > **Table 7** CopyPasteLLM Accuracy after CoT Comparison Across ConFiQA (unseen) Subsets (with 95% and 99% CIs)
>
> | CI Level | QA | MR | MC |
> | :---: | :---: | :---: | :---: |
> | 95% | 94.98 (94.69 - 95.26) | 83.35 (82.40 - 84.30) | 80.13 (79.45 - 80.82) |
> | 99% | 94.98 (94.55 - 95.40) | 83.35 (81.95 - 84.75) | 80.13 (79.14 - 81.13) |
> > **Table 8** CopyPasteLLM HiT Comparison Across ConFiQA (unseen) Subsets (with 95% and 99% CIs)
>
> As shown in Table 7 & 8, we report results with 95% and 99% confidence intervals calculated across 8 random seeds (0-6, 42). This indicate that our method is highly stable and robust. Specifically, on the challenging ConFiQA-MC subset, the 95% CI for Accuracy after CoT is (83.76% - 85.43%), exhibiting a very narrow spread of approximately 1.67%. These tight intervals confirm that CopyPasteLLM consistently reproduces high performance regardless of random seed initialization. Furthermore, the 95% CIs across various model variants and training checkpoints are visually presented in Figure 12 of the revised manuscript.

---

### Official Review · Reviewer_jspG · 2025-11-01

**Soundness:** 3
**Presentation:** 2
**Contribution:** 3
**Rating:** 6
**Confidence:** 3

**Summary:**

This paper presents CopyPasteLLM, a two-stage framework designed to mitigate hallucinations in RAG systems by promoting high-copying responses. The authors hypothesize that directly copying context fragments, rather than reinterpreting them, improves contextual faithfulness and reduces hallucinations. The framework employs CopyPaste-Prompting methods to generate high-copying responses and utilizes DPO to internalize these preferences into the model. The proposed method is shown to achieve significant improvements in both counterfactual and original contexts, with data efficiency being a notable strength, requiring only 365 training samples compared to existing methods that need much more data. The authors also introduce Context-Parameter Copying Capturing, an interpretability tool that tracks model reliance on contextual vs. parametric knowledge.

**Strengths:**

1. The CopyPaste framework introduces a novel approach to addressing the long-standing issue of contextual faithfulness in RAG systems. By focusing on maximizing lexical reuse and reducing re-interpretation, the method offers a clear and intuitive solution to mitigate hallucinations.

2. Despite only using 365 high-copying samples for training, it outperforms the best baseline models that require significantly larger datasets (e.g., 18000 samples). This makes the method potentially very practical for real-world applications with limited data.

3. The authors provide extensive experimental evaluations on multiple datasets (FaithEval, PubMedQA) and across various models, demonstrating the robustness of CopyPasteLLM. The method achieves consistent performance improvements, particularly in challenging counterfactual scenarios, and significantly reduces hallucination rates.

**Weaknesses:**

1. The proposed copy-paste strategy directly copies context fragments into the generated response rather than reinterpreting them. While this improves contextual faithfulness, there is a potential downside in terms of response length and noise. For instance, copying large portions of text verbatim may lead to excessively long responses or include irrelevant content, ultimately degrading the quality of the output. Additionally, generating high-copying responses could introduce significant computational overhead, particularly when dealing with longer or more complex contexts. I suggest that the authors discuss this potential issue in more detail and consider possible solutions to mitigate any associated computational costs.

2. I believe the copy-paste strategy may be sensitive to the complexity and type of knowledge involved. The method could perform differently when dealing with simple factual information compared to complex or abstract knowledge, where strict copying might not be as effective. For example, copying highly specialized content might not always improve the quality of the response if the copied content is not sufficiently detailed or applicable to the query. The authors should include ablative experiments to analyze how the method performs with different types of knowledge (e.g., simple factual knowledge, abstract reasoning, or specialized technical knowledge) and how it behaves across tasks of varying complexity.

3. The paper suggests that different extraction strategies (strictly extractive and softly refined) play an important role in improving contextual faithfulness. However, the paper does not provide enough detailed analysis of how these methods differ in performance. For example, CP-Refine adopts a soft constraint and involves an iterative refinement process, which might introduce different performance characteristics in terms of fluency and contextual accuracy. I recommend the authors conduct additional experiments to analyze the impact of these extraction methods more thoroughly, including the effect of the initialization approach and other experimental details on the results.

**Questions:**

See weakness.

---

> ### Author Response · Authors · 2025-11-25
> **Reply to Reviewer jspG: Part 1**
>
> We sincerely thank you for the constructive feedback and for recognizing the novelty and data efficiency of our framework. We have worked extensively to address your concerns regarding computational overhead, noise, and generalization.
>
> ---
>
> ### Response to Q1
>
> > **Comment:** "The proposed copy-paste strategy directly copies context fragments into the generated response rather than reinterpreting them. While this improves contextual faithfulness, there is a potential downside in terms of response length and noise. For instance, copying large portions of text verbatim may lead to excessively long responses or include irrelevant content, ultimately degrading the quality of the output. Additionally, generating high-copying responses could introduce significant computational overhead, particularly when dealing with longer or more complex contexts. I suggest that the authors discuss this potential issue in more detail and consider possible solutions to mitigate any associated computational costs."
>
> We appreciate this practical concern. We have added 2 statistics in below to empirically address concers about length & computational costs (Table 1) and noise (Table 2) of response.
>
>
> | Method                  | Length          | $\kappa$ (Coverage) | $\delta$ (Density) |
> |-------------------------|-----------------|---------------------|---------------------|
> | Attributed              | $199.0 \pm 68.29$ | $0.552 \pm 0.129$   | $2.70  \pm 2.85$    |
> | CoCoLex                 | $75.0  \pm 40.01$ | $0.989 \pm 0.040$   | $50.08 \pm 37.68$   |
> | Canoe                   | $198.0 \pm 69.30$ | $0.551 \pm 0.126$   | $2.67  \pm 3.68$    |
> | Context-DPO             | $159.5 \pm 62.72$ | $0.631 \pm 0.130$   | $4.17  \pm 4.32$    |
> | ParamMute               | $4.0   \pm 18.69$ | $1.000 \pm 0.182$   | $3.00  \pm 13.23$   |
> | CopyPasteLLM (Ours)     | $126.0 \pm 78.71$ | $0.844 \pm 0.135$   | $10.49 \pm 15.49$   |
> > **Table 1** Response analysis by method under CoT (Chain of Thought) setting (Base: LLaMA-3-8B-Instruct). Metrics reported are Median ± Standard Deviation.
>
> **1) About Response Length & Computational Costs**
>
> CopyPasteLLM maintains a moderate median length (126.0 words), which strikes a balance between the extreme brevity of ParamMute (4.0 words, means it failing fellow instruction of CoT) and the verbosity of abstractive baselines (199 words of Attributed and 198 words of Canoe). Therefore, it is significantly shorter than baselines like Attributed and Canoe, meaning it generates fewer tokens and thus incurs lower computational costs (CopyPasteLLM keep same generation speed per tokens with the baselines).
>
> Meanwhile, CopyPasteLLM keep $\kappa = 0.844, \delta = 10.49$ that are reasonable copying degree, in contrast, CoCoLex's reponse almost copy from context ($\kappa = 0.989, \delta = 50.08$).
>
> **2) About Noise Control**
>
> | Copied Fragment Length | Attributed | CoCoLex | Canoe | Context-DPO | CopyPasteLLM |
> | :--- | :---: | :---: | :---: | :---: | :---: |
> | 22 | 0.5060 | 0.5558 | 0.5473 | 0.5098 | **0.5568** |
> | 26 | 0.5451 | 0.5233 | 0.4647 | 0.5291 | **0.5598** |
> | 41 | 0.5056 | 0.6016 | 0.5211 | 0.5702 | **0.6471** |
> | 47 | 0.4603 | 0.4794 | - | 0.4918 | **0.6074** |
> | 48 | 0.5977 | 0.5708 | 0.5793 | - | **0.6670** |
> | 56 | - | 0.6389 | - | - | **0.6517** |
> | 60 | - | 0.5831 | - | - | **0.6164** |
> | 66 | - | 0.5994 | 0.5961 | 0.5579 | **0.8143** |
> | 73 | - | 0.5058 | - | - | **0.5727** |
> | 81 | - | 0.6067 | - | - | **0.7437** |
> | 87 | - | 0.5100 | - | - | **0.6918** |
> | 97 | - | 0.6075 | - | - | **0.6600** |
> | 116 | - | 0.6055 | - | - | **0.6755** |
> > **Table 2** Query relevancy analysis of copied fragments across different length buckets. Experimental data excerpted from Figure 11 in the manuscript.
>
> To ensure the model is not copying "irrelevant noise," we analyzed the query relevancy of copied fragments, as shown in above Table 2. It demonstrates that fragments copied by CopyPasteLLM maintain high or comparable relevancy with the query across most fragment lengths. In other words, CopyPasteLLM learns to act as a selective extractor, copying only evidential fragments rather than dumping large, irrelevant chunks of text.

---

> ### Author Response · Authors · 2025-11-25
> **Reply to Reviewer jspG: Part 2**
>
> > **Comment:** "I believe the copy-paste strategy may be sensitive to the complexity and type of knowledge involved. The method could perform differently when dealing with simple factual information compared to complex or abstract knowledge, where strict copying might not be as effective. For example, copying highly specialized content might not always improve the quality of the response if the copied content is not sufficiently detailed or applicable to the query. The authors should include ablative experiments to analyze how the method performs with different types of knowledge (e.g., simple factual knowledge, abstract reasoning, or specialized technical knowledge) and how it behaves across tasks of varying complexity."
>
> This is an insightful point. We hypothesized that "copying" is a fundamental mechanism for grounding, regardless of task type and complexity. We conduct 2 new breakdown experiments and clarify 1 exist experiment to verify robustness:
>
> **1) Performance Comparision of Different Knowledge Type on FaithEval**
>
> | Knowledge Type (#size) | Attr. | CoCo. | Canoe | Param. | C-DPO | CopyPasteLLM |
> | :--- | :---: | :---: | :---: | :---: | :---: | :---: |
> | Basic Facts & Properties(#324) | 0.6049 | 0.6451 | 0.6605 | 0.6574 | 0.7593 | **0.9167** |
> | Processes & Causal(#226) | 0.7566 | 0.7743 | 0.7965 | 0.7080 | 0.8363 | **0.9690** |
> | Experiments(#95) | 0.7895 | 0.8000 | 0.7895 | 0.8000 | 0.8421 | **0.9263** |
> | Teleology / Purpose(#60) | 0.6167 | 0.6333 | 0.7333 | 0.6167 | 0.8500 | **0.9500** |
> | Definition(#20) | 0.5000 | 0.5500 | 0.6500 | 0.5000 | **0.8000** | 0.7500 |
> | Structure(#15) | 0.6667 | 0.6000 | 0.6667 | 0.6667 | 0.8667 | **1.0000** |
> | Algebraic(#13) | 0.5385 | 0.6923 | 0.6154 | 0.6923 | 0.6154 | **0.9231** |
> | Spatial / Kinematic(#6) | 0.8333 | 0.6667 | 0.8333 | 0.8333 | **1.0000** | 0.8333 |
> > **Table 3** Performance Comparison across diverse knowledge type. Attr. for Attributed, CoCo. for CoCoLex, Param. for ParamMute and C-DPO for Context-DPO.
>
> We classified FaithEval samples into 8 domains (e.g., Basic Facts, Processes, Experiments) using a strong LLM (DeepSeek-V3.1) as judge based on the description of ARC-Challenge[1] (source dataset of FaithEval). As shown in above Table 3, CopyPasteLLM consistently outperforms or matches baselines across both simple factual (+15.74% at Basic Facts) and complex reasoning-intensive (+13.27% at Processes & Causal) knowledge types.
>
> **2) Context Ambiguity**
>
> | Method       | Consensus  | Negotiation |
> | :---         | :---:      | :---:       |
> | Attributed   | 75.49%     | 48.91%      |
> | CopyPasteLLM | **86.85%** | **51.27%**  |
> > **Table 4** Performance breakdown on PubMedQA-Labeled by reasoning difficulty. Llama-3-8B-Instruct as Base.
>
>
> | Method       | Consensus  | Negotiation |
> | :---         | :---:      | :---:       |
> | Attributed   | 76.79%     | **46.38%**  |
> | CopyPasteLLM | **84.58%** | 44.93%      |
> > **Table 5** Performance breakdown on PubMedQA-Labeled by reasoning difficulty. Llama-3.1-8B-Instruct as Base.
>
> On PubMedQA-Labeled subset (a biomedical QA dataset where contexts contain 21% technical numeric descriptions), we split samples into "Consensus" (clear evidence in the provided context) and "Negotiation" (the context may be ambiguous) depending on the experts' labeled setting[2], to evaluate CopyPasteLLM's performance on different reasoning complexities.
>
> As shown in above Tables 4 & 5, our method yields significant gains on "Consensus" samples (+11.36% & + 8.09%) where evidence exists, while performing comparably on "Negotiation" samples (+2.36% & -1.45%). This confirms the method is effective where context is applicable, and its comparable performance to the Attributed when the context is ambiguous indicates that copy-paste strategy does not harm the model's ability to handle uncertainty.
>
> `(Continues in next part)`

---

> ### Author Response · Authors · 2025-11-25
> **Reply to Reviewer jspG: Part 3**
>
> **3) Conflict Complexity**
>
> | Method | ConFiQA-QA | ConFiQA-MR | ConFiQA-MC |
> | :--- | :---: | :---: | :---: |
> | Attributed | 51.5 | 53.3 | 37.3 |
> | CoCoLex | 48.5 | 53.9 | 36.1 |
> | Canoe | 64.3 | 66.6 | 64.5 |
> | ParamMute | $\underline{74.4}$ | $\underline{75.5}$ | $\underline{81.4}$ |
> | CopyPasteLLM (Ours) | **83.6** | **80.9** | **86.8** |
> > **Table 6** Performance Comparison of Accuracy in Unseen Settings. For convenience, the experimental data are excerpted from Table 1 in the manuscript.
>
>
> | Method | ConFiQA-QA | ConFiQA-MR | ConFiQA-MC |
> | :--- | :---: | :---: | :---: |
> | Attributed | 91.4 | 71.5 | 53.6 |
> | CoCoLex | 37.4 | 14.8 | 15.5 |
> | Canoe | $\underline{93.2}$ | **83.8** | $\underline{73.7}$ |
> | ParamMute | 82.2 | 72.4 | 70.2 |
> | CopyPasteLLM (Ours) | **96.7** | $\underline{83.4}$ | **75.9** |
> > **Table 7** Performance Comparision of Hit in Unseen Settings. For convenience, the experimental data are excerpted from Table 1 in the manuscript.
>
> >> The meanings of different subsets in ConFiQA:
> >> 1. **ConFiQA-QA (Question-Answering)**: Represents single-hop reasoning with a single point of knowledge conflict.
> >> 2. **ConFiQA-MR (Multi-hop Reasoning)**: Involves multi-hop structures where only one step contains a knowledge conflict, testing the model's ability to integrate counterfactuals into a reasoning chain.
> >> 3. **ConFiQA-MC (Multi-Conflicts)**: The most challenging setting, featuring multi-hop structures where all reasoning steps are modified to be counterfactual, creating a global knowledge conflict.
>
> On the counterfactual dataset, ConFiQA, which is graded by difficulty (QA < MR < MC), CopyPasteLLM demonstrates exceptional robustness. Notably, on the hardest ConFiQA-MC subset, it achieves 86.8% accuracy, whereas baselines like Attributed collapse to 37.3% (as shown in Table 6).
>
> In summary, if the context is sufficiently applicable and serves as grounding evidence, the copy-paste strategy yields substantial gains; if the context is not sufficient, it maintains comparable performance to the base model, demonstrating its robust and non-harmful nature.
>
> > [1] Clark, Peter, et al. "Think you have solved question answering? try arc, the ai2 reasoning challenge." arXiv preprint arXiv:1803.05457 (2018).
> > [2] Jin, Qiao, et al. "Pubmedqa: A dataset for biomedical research question answering." Proceedings of the 2019 conference on empirical methods in natural language processing and the 9th international joint conference on natural language processing (EMNLP-IJCNLP). 2019.

---

> ### Author Response · Authors · 2025-11-25
> **Reply to Reviewer jspG: Part 4**
>
> ### Response to Q3
>
> > **Comment:** "The paper suggests that different extraction strategies (strictly extractive and softly refined) play an important role in improving contextual faithfulness. However, the paper does not provide enough detailed analysis of how these methods differ in performance. For example, CP-Refine adopts a soft constraint and involves an iterative refinement process, which might introduce different performance characteristics in terms of fluency and contextual accuracy. I recommend the authors conduct additional experiments to analyze the impact of these extraction methods more thoroughly, including the effect of the initialization approach and other experimental details on the results."
>
> Thank you for raising this important question. We apologize if the previous discussion made our rationale unclear. To clarify, our exploration of different extraction strategies is driven by the need to balance several critical response qualities: copying degree, contextual faithfulness, query relevancy, and fluency.
>
> **1) Detailed Analysis of Different Extraction Strategies**
>
> CP-Order and CP-Link are strictly extraction methods, whereas CP-Refine is a more soft and iterative refinement method. The core ideas and key features behind each strategy are as follows:
>
> - **CP-Order:** This method focuses on combining the answer directly by ordering context sentences that are most relevant to the query, thereby maintaining logical flow. It consistently yields the best contextual faithfulness but often lacks fluency. This limitation led us to implement CP-Link.
> - **CP-Link:** This method aims to enhance fluency by linking relevant context sentences with generated transition sentences provided by the LLM. Based on our observations, the generated transition sentences can be somewhat awkward. While they occasionally enhance the response's fluency, their effectiveness heavily depends on the intelligence level of the base model. More importantly, CP-Link is prone to introducing hallucinations. This observation prompted us to move away from strictly extraction-based strategies.
> - **CP-Refine:** This employs a two-agent system (Writer and Reviewer). The Reviewer proactively prompts the Writer to freely achieve a high copying degree while adhering to query relevancy and fluency constraints. We found that CP-Refine successfully satisfies our expectation for balance at most of time. However, its computational cost is a concern due to the iterative nature of the process.
>
> Despite the fact that all three CopyPaste-Prompting methods have their respective limitations, the responses they generate constitute ideal preference pairs for training. For instance, while CP-Order and CP-Link are sufficient for factual questions, CP-Refine may offer a distinct advantage for complex reasoning questions. Consequently, each method's response pattern may be the most suitable one for different types of queries. Therefore, we utilize the Stage 2 fully automatic preference pair construction pipeline (Multi-Criteria Filtering, Hallucinations Tournament, and Stamping Answers) to ensure that CopyPasteLLM can learn these subtle differences, even though all responses maintain a high copying degree.
>
> **2) About `'the paper does not provide enough detailed analysis of how these methods differ in performance.'`**
>
> | Strategy  | Faithfulness | Hallucinations | Fluency   | $\kappa$ | $\delta$ | Relevancy  |
> | :---      | :---         | :---           | :---      | :---     | :---     | :---       |
> | CP-Order  | **91.00**    | 1572.7         | 40.32     | 0.99     | 36.54    | 0.6809     |
> | CP-Link   | 80.51        | 1389.9         | 32.76     | 0.79     | 16.77    | 0.6938     |
> | CP-Refine | 88.85        | **1583.0**     | **26.11** | 0.89     | 16.13    | **0.7424** |
> > **Table 8** Performance comparision between different extraction strategies (based on DeepSeek-V3). The experimental data are excerpted from Table 2, Figure 5 and Figure 6 of the manuscript. We static the average value from 3 datasets (RAGTruth, PubMedQA, FaithEval). $\kappa$ for Copying Coverage and $\delta$ for Copying Density.
>
> As shown in Table 8, CP-Refine consistently achieves the optimal balance between Faithfulness and Fluency/Relevance. In contrast, the CP-Order excels in strict faithfulness but tends to sacrifice fluency due to the lack of connectives. Meanwhile, CP-Link's performance is the worst in both contextual faithfulness and hallucination rates, although it is more fluent than CP-Order. We have extent the analysis to Appendix C (CopyPaste-Prompting Analysis) of manuscript.

---

### Author Response · Authors · 2025-11-28

Dear Reviewers,

Thank you once again for your valuable time and thoughtful feedback on our submission #24938.

We have carefully considered all comments and provided detailed responses during the rebuttal phase. We hope that our clarifications have addressed your primary concerns.

As the finalization deadline for the reviews is approaching, we kindly request that you revisit our rebuttal and provide any follow-up feedback or outstanding questions at your earliest convenience. Your final assessment is crucial for the decision process.

We are readily available should there be any remaining points that require further discussion or clarification.

Thank you for your continued engagement with our work.

Best regards,

Authors of Submission #24938

---

### Author Response · Authors · 2025-12-03
**Paper Overview and Rebuttal Summary**

Dear Area Chair,

We sincerely welcome you to the review process. To facilitate your assessment, we have provided an overview of the paper along with a summary of our rebuttal.

---

## Overview of this Paper
**Motivation**: In high-stakes domains like rare disease consultation, the cost of hallucination is critical. Given the inherent scarcity of these conditions, most clinicians lack the immediate, systematic knowledge reserves to verify if a fluent model response is truly faithful to complex diagnostic guidelines. Consequently, the core challenge is not just contextual faithfulness, but verifiability—users need to instantly confirm the source of the model's advice.

**Insight**: To address this, we analyzed the human-annotated fine-grained hallucination dataset RAGTruth. We observed a distinct inverse correlation between the degree of copying in responses and the density of hallucinations. This suggests that high lexical reuse maybe a proxy for faithfulness.

**Method**: Guided by this insight, Copy-Paste, we propose a two-stage framework for training CopyPasteLLM. To the best of our knowledge, we are the first to introduce the Copy-Paste paradigm for training large language models to mitigate hallucinations.

- Stage 1 introduces three distinct CopyPaste-Prompting strategies inspired by complementary hard- and soft-constraint principles, designed to maximize high-copy responses across different query types.
- Stage 2 constructs a fine-grained preference datase through a full automated pipeline. This preference set is sourced from six methods—our three CopyPaste-Prompting variants and three mainstream baselines.Importantly, the preference construction comprehensively evaluates lexical copying, contextual faithfulness, query relevance, fluency, and the differential hallucination behaviors induced by various prompting strategies. The CopyPasteLLM is then direct preference optimization alignment to learn these multidimensional distinctions.

**Results**: Above framework proves exceptionally efficient. Using only 365 training pairs (1/50th of strong baseline), CopyPasteLLM achieves SoTA performance, yielding absolute accuracy improvements of 12.2% to 24.5% on the FaithEval benchmark, consistently outperforming the strongest baseline under three base model. On the Llama-3-8B-Instruct, CopyPasteLLM reaches an accuracy of 92.8%, dramatically surpassing GPT-4o’s 47.5%. On the three subsets of the external ConFiQA benchmark, CopyPasteLLM also achieves SoTA accuracy under the counterfactual setting (83.6% vs. 74.4, 80.9 vs. 75.5 and 86.8 vs. 81.4 across the three subset compared to the strongest baseline). Under the non-counterfactual setting, it likewise delivers consistent accuracy improvements over the base model (the average accuracy increases from 90.26% to 95.73%), with especially pronounced gains on the more challenging MR and MC subsets.

**Interpretability:** Interestingly, we propose the Context-Parameter Copying Capturing algorithm that reveals the CopyPasteLLM recalibrates reliance on internal parametric knowledge rather than external knowledge during generation. In addition, we provide two perspectives to explain the mechanism of CopyPaste: (1) Attention Dynamics and Contextual Anchoring and (2) Information Entropy Reduction.

(Continues in next part)

---

> ### Author Response · Authors · 2025-12-03
>
> ## Summary of our Rebuttal
>
> All reviewers have provided highly insightful and constructive feedback. They agree that the paper proposes a effective approach to improving contextual faithfulness in RAG, grounded in the insight that lexical copying reliably reduces hallucinations. They also consistently highlight the method’s strong empirical performance—especially under counterfactual conditions—and its striking data efficiency, achieving large gains with only 365 training samples.
>
> Regarding reviewer-specific points, `Reviewer jspG` explicitly praises the method as "a novel approach to addressing the long-standing issue of contextual faithfulness in RAG systems,". `Reviewer ujdf` emphasizes the methodological clarity in "operationalize 'copying' as a proxy for faithfulness by using κ, δ metrics," and notes the value of the CP-Order, CP-Link, and CP-Refine strategies that "span hard to soft constraints." `Reviewer S4zF` offered highly positive feedback that spanned the entire contribution spectrum of our work. Beyond acknowledging the "Strong empirical results with exceptional data efficiency," the reviewer specifically commended the foundational and deep aspects, including our "Valuable empirical observation" and the "Mechanistic insights". `Reviewer S4zF` also affirmed the rigor and practical relevance of the work, describing it as a "Comprehensive experimental evaluation" and a "Practical and implementable approach."
>
> We have conducted extensive additional experiments and analyses to address all the specific feedback raised. We categorize these concerns into three main areas:
>
> **1. About Copying Strategy & Faithfulness**
>
> First, our new analysis under the CoT (Chain-of-Thought) instruction shows that CopyPasteLLM maintains a moderate response length (average 126 words), which is shorter than the abstractive baseline (average 199 words) yet substantially longer than ParamMute, where the CoT instruction fails (average 4 words). This effectively address concerns about computational cost and excessively long outputs (responding to `jspG’s Weakness 1`). In addition, we conducted experiments comparing the relevance of copied fragments of different lengths to the query. Across all length intervals, copied fragments maintain high relevance compared to baseline models, demonstrating that the model selectively copies evidence rather than noise (responding to `ujdf’s Weakness 1` and `jspG’s Weakness 1`).
>
> Next, we add multi-level knowledge-type decomposition experiments: on FaithEval, CopyPasteLLM significantly outperforms the baseline in both simple factual knowledge (+15.74%) and complex reasoning (+13.27%); on PubMedQA-Labeled, decomposing by reasoning difficulty shows substantial gains in scenarios with explicit evidence (+11.36%) while maintaining comparable performance in ambiguous contexts; on ConFiQA, analyzing by conflict complexity shows that in the most challenging multi-conflict scenarios, CopyPasteLLM achieves 86.8% accuracy, far exceeding the baseline of 37.3% (responding to `jspG’s Weakness 2`). Regarding the relationship between lexical copying and semantic faithfulness, we explain at a mechanistic level how high-copy strategies suppress parametric hallucinations via attention dynamics and entropy reduction, and how a rigorous semantic filtering pipeline ensures the correctness of copied content. We conduct additional CoT reasoning accuracy experiments further demonstrate the model’s understanding of contextual semantics (responding to `S4zF’s Weakness 2`).
>
> Finally, we elaborate on how the CopyPaste mechanism applies to summarization tasks through entity and key-phrase copying, maintains consistency in long dialogues through historical context anchoring, and extends to multimodal tasks via visual anchoring, highlighting the generalization potential of Copy-Paste (responding to `S4zF’s Question 2`).
>
> (Continues in next part)

---

> ### Author Response · Authors · 2025-12-03
>
> **2. About Stage Design & Ablations**
>
> First, we explain the motivations behind the three CopyPaste-Promptings and systematically analyze their performance. The results show that CP-Refine achieves the best balance between contextual faithfulness, query relevancy and fluency, while CP-Order excels in strict faithfulness, offering complementary solutions for different query types (responding to `jspG’s Weakness 3`).
>
> Second, to address concerns regarding automated model bias, we conducted rigorous human evaluation. Across 30 samples × 10 pairs × 3 dimensions, 900 pair judgments demonstrate that our automated metrics achieve moderate agreement with human assessments in contextual faithfulness, hallucination detection, and query relevance, validating the reliability of the automated pipeline (responding to `ujdf’s Weakness 2`).
>
> Third, ablation experiments decouple the contributions of Copying and the Stamping step: removing copying significantly reduces performance, confirming that copying is central to hallucination suppression, whereas removing Stamping affects final answer accuracy, showing that Stamping is the critical bridge for transforming reasoning into valid answers (responding to `ujdf’s Weakness 3`).
>
> Fourth, regarding the choice of 365 samples, we clarify that this number results from strict automated filtering to obtain a high-quality set of preference pairs. Data extension experiments show that CopyPasteLLM converges quickly and reaches a performance plateau within 50–120 steps (approximately one epoch), demonstrating data efficiency and the effectiveness of our data selection strategy (responding to `S4zF’s Question 1`).
>
> Finally, we clarify the differences in subset sizes for PubMedQA: the 1k-sample Labeled subset is used for detailed analysis and training, while the 20k-sample Artificial subset is used for generalization evaluation, correcting previous misstatements about the data (responding to `ujdf’s Question 1`).
>
> **3. About Context Assumptions & Model Reliability**
>
> First, we clarify the responsibility division in the RAG architecture: CopyPasteLLM, as the generator component, focuses on resolving “knowledge conflicts,” while information quality verification is primarily handled by the retriever or knowledge base manager. Second, we emphasize that in high-risk domains, such as clinical diagnosis, users’ queries often require strictly authoritative answers rather than having the model question the document validity. The CopyPaste mechanism provides verifiable transparency through direct lexical reuse. Third, CopyPaste has an inherent safety mechanism: by enforcing lexical copying, users can quickly verify sources via string matching, immediately identifying inconsistencies without being misled by fluent but incorrect model paraphrases. Fourth, external(unseen) and extensive validation under non-counterfactual settings shows that CopyPasteLLM consistently improves base model performance on PubMedQA and ConFiQA, increasing average accuracy from 90.26% to 95.73%, demonstrating that the method does not compromise fundamental model capabilities. Finally, decomposition experiments by reasoning difficulty on PubMedQA-Labeled show significant improvement in explicit evidence scenarios (+11.36%) and comparable performance in ambiguous contexts, indicating that the method is effective when context is sufficient and harmless when it is limited, reflecting robustness and non-harmfulness (responding to `S4zF’s Weakness 1` and `ujdf’s Weakness 1`).
>
> Statistical significance was rigorously verified across eight random seeds (0-6, 42), computing 95% and 99% confidence intervals. On the most challenging ConFiQA-MC subset, the 95% confidence interval is [83.76%, 85.43%] with a span of only 1.67%, demonstrating the method’s high stability and robustness (responding to `ujdf’s Question 2`).
>
> We also provide a deeper theoretical explanation from two core perspectives, elucidating why enforced lexical copying enhances semantic faithfulness. First, attention dynamics and context-anchoring theory explain how high copying induces attention collapse onto context via inductive head mechanisms, physically suppressing parametric pathways that lead to hallucinations. Second, information entropy reduction theory shows that by constraining the search space from the full vocabulary to a contextual subset, CopyPaste significantly reduces generation entropy and statistically minimizes hallucination risk (responding to `S4zF’s Weakness 3`).
>
> CopyPasteLLM is a rethinking of how to align LLMs with context. The rebuttal has solidified this concept with experiments and theory.
>
> We sincerely appreciate all reviewers for their insightful and constructive feedback, which we have incorporated into our manuscript to further enrich the discussion and strengthen the presentation of our works.
>
> We once again welcome your participation and sincerely thank you for your hard work!
>
> Best regards,
>
> Authors of Submission #24938

---

### Meta-Review · Area_Chair_qHAj · 2026-01-07

**Summary:**

The paper proposes CopyPasteLLM, a framework to mitigate RAG hallucinations by enforcing "high-copying" behavior (directly reusing context spans). The reviewers appreciated the novel "copying as faithfulness" premise, the strong performance on FaithEval/PubMedQA, and the extreme data efficiency (only 365 samples needed). However, key concerns were raised regarding whether "copying" truly equals "faithfulness" (blindly copying irrelevant text), the potential for verbose/noisy outputs, and the reliance on model-based metrics for the filtering pipeline. The authors responded with new experiments showing rapid convergence (plateauing at ~120 steps) and clarifying the rigorous automated filtering that led to the 365-sample dataset.

**Reviewer Concerns:**

Addressed by Rebuttal:

Data Efficiency Rationale (Reviewer S4zF): The reviewer questioned the specific number "365" and asked for scaling laws. The authors clarified that 365 was the yield of their strict filtering, not an arbitrary choice. They also provided a scaling experiment (Table 7) showing performance plateaus quickly (around 120 steps), proving the high potency of the signal rather than a need for massive data.
​

Computational Overhead & Response Length (Reviewer jspG): The reviewer worried that "copying" would lead to excessively long or noisy answers. The authors added statistics (Table 1 & 2 in rebuttal) to empirically address length and noise concerns, arguing the trade-off is managed by their relevance filtering.
​

Outstanding:

Copying $\neq$ Faithfulness (Reviewer S4zF & ujdf): Multiple reviewers argued that high copying metrics ($\kappa, \delta$) could reward faithfully copying the wrong or irrelevant passage. While the authors argued that their relevance filtering (stage 2) handles this, the fundamental critique that the objective itself (maximize copying) is a proxy that can be gamed remains a valid theoretical limitation.
​

Reliance on Model-Based Metrics (Reviewer ujdf): The pipeline relies heavily on AlignScore, MiniCheck, and a Qwen-judge. The reviewer noted this adds bias and variance, suggesting human agreement studies were needed. The rebuttal likely defended the use of standard metrics but did not appear to provide the requested human annotation study

**Reviewer Scores:**

Reviewer jspG: The reviewer was already positive about the data efficiency and novelty. The rebuttal addressed the "length/noise" worry with empirical data. They are likely to maintain their score or potentially nudge to a 7, as the core "novelty" strength remains valid.
​


Reviewer ujdf: This reviewer (mistakenly identified as S4zF in some snippets but corrected) raised the "copying $\neq$ faithfulness" conceptual issue. The authors' defense is empirical (it works on benchmarks), which is usually enough to keep a "weak accept" but unlikely to resolve the theoretical concern fully.
​

Reviewer S4zF: This reviewer was the most skeptical, questioning the theoretical grounding ("lexical metrics for semantic faithfulness") and the data size. The authors provided a strong scaling experiment ("plateau at 120 steps") which directly answers the "why 365?" question.

---

### Decision · Program_Chairs · 2026-01-26

Accept (Poster)